# Excessive miR-25-3p maturation via $N^6$-methyladenosine stimulated by cigarette smoke promotes pancreatic cancer progression

Jialiang Zhang[1,8], Ruihong Bai[1,8], Mei Li[2,8], Huilin Ye[3], Chen Wu[4,5], Chengfeng Wang[6], Shengping Li[1], Liping Tan[1], Dongmei Mai[1], Guolin Li[3], Ling Pan[1], Yanfen Zheng[1], Jiachun Su[1], Ying Ye[1], Zhiqiang Fu[3,7], Shangyou Zheng[3,7], Zhixiang Zuo[1], Zexian Liu[1], Qi Zhao[1], Xu Che[6], Dan Xie[1], Weihua Jia[1], Mu-Sheng Zeng[1], Wen Tan[4,5], Rufu Chen[3], Rui-Hua Xu[1], Jian Zheng[1] & Dongxin Lin[1,4,5]

$N^6$-methyladenosine (m6A) modification is an important mechanism in miRNA processing and maturation, but the role of its aberrant regulation in human diseases remained unclear. Here, we demonstrate that oncogenic primary microRNA-25 (miR-25) in pancreatic duct epithelial cells can be excessively matured by cigarette smoke condensate (CSC) via enhanced m6A modification that is mediated by NF-κB associated protein (NKAP). This modification is catalyzed by overexpressed methyltransferase-like 3 (METTL3) due to hypomethylation of the *METTL3* promoter also caused by CSC. Mature miR-25, miR-25-3p, suppresses PH domain leucine-rich repeat protein phosphatase 2 (PHLPP2), resulting in the activation of oncogenic AKT-p70S6K signaling, which provokes malignant phenotypes of pancreatic cancer cells. High levels of miR-25-3p are detected in smokers and in pancreatic cancers tissues that are correlated with poor prognosis of pancreatic cancer patients. These results collectively indicate that cigarette smoke-induced miR-25-3p excessive maturation via m6A modification promotes the development and progression of pancreatic cancer.

[1] Sun Yat-sen University Cancer Center, State Key Laboratory of Oncology in South China and Collaborative Innovation Center for Cancer Medicine, Guangzhou, China. [2] Department of Pathology, Sun Yat-sen University Cancer Center, Guangzhou, China. [3] Department of Pancreaticobiliary Surgery, Sun Yat-sen Memorial Hospital, Sun Yat-sen University, Guangzhou, China. [4] Department of Etiology and Carcinogenesis, National Cancer Center/National Clinical Research Center/Cancer Hospital, Chinese Academy of Medical Sciences and Peking Union Medical College, Beijing, China. [5] CAMS Key Laboratory of Genetics and Genomic Biology, Chinese Academy of Medical Sciences and Peking Union Medical College, Beijing, China. [6] Department of Abdominal Surgery, National Cancer Center/National Clinical Research Center/Cancer Hospital, Chinese Academy of Medical Sciences and Peking Union Medical College, Beijing, China. [7] Guangdong Provincial Key Laboratory of Malignant Tumor Epigenetics and Gene Regulation, Sun Yat-sen Memorial Hospital, Sun Yat-sen University, Guangzhou, China. [8] These authors contributed equally: Jialiang Zhang, Ruihong Bai, Mei Li. Correspondence and requests for materials should be addressed to R.C. (email: chenrf63@163.com) or to R.-H.X. (email: xurh@sysucc.org.cn) or to J.Z. (email: zhengjian@sysucc.org.cn) or to D.L. (email: lindx@sysucc.org.cn)

Pancreatic cancer, mostly pancreatic ductal adenocarcinoma (PDAC), is a lethal malignancy and the 5-year survival rates are usually <5%[1,2]. The initiation and progression of PDAC is believed to be attributed to a gene–environment interaction[3]. As one of the established environmental risk factors for PDAC, tobacco smoking can accelerate chronic pancreatitis and trigger pancreatic carcinogenesis[4–7]. Smokers generally display a PDAC risk two- to threefold higher than nonsmokers and the risk increases with the extent and the time of exposure[8–10]. However, the underlying mechanism has yet to be elucidated. Unlike other smoking-related cancers such as lung cancer, tobacco smoking does not seem to promote PDAC by mutating the known driver genes such as *TP53* and *KRAS*[11,12].

Tobacco smoking can cause the aberrant expression of microRNA (miRNA) encoding genes[13–15]. Since miRNAs widely participate in the spatiotemporal regulation of mRNA and protein synthesis[16,17], their aberrant expression could lead to certain diseases including the initiation and progression of malignancies[18,19]. In cells, miRNAs are initially transcribed as primary miRNAs (pri-miRNAs), which are then cut into hairpin-structured precursor miRNAs (pre-miRNAs) by the nuclear microprocessor complex containing RNase III enzyme RNASEN (Drosha) and DiGeorge critical region 8 (DGCR8). Pre-miRNAs are processed to form mature miRNAs (usually 22−25 nucleotides) that actually perform the functions[20]. Although this process has long been recognized, why and how it is perturbed in human cancers are still unclear.

$N^6$-methyladenosine (m$^6$A) is the most prevalent internal chemical modification of RNAs in eukaryotes[21,22] and m$^6$A on pri-miRNAs can perturb miRNA maturation[23,24]. The deposit of m$^6$A requires (a) a writer protein complex consisting of methyltransferase-like 3 (METTL3), methyltransferase-like 14 (METTL14), and Wilms tumor 1-associated protein (WTAP)[25,26], and (b) a reader protein complex recognizing and processing the modified pri-miRNAs to produce mature miRNAs that affect cellular processes[24]. The readers are expected to be miRNA-specific[24] and have not been fully identified.

Here we report the oncogenic role of miR-25-3p in pancreatic cells induced by cigarette smoke condensate (CSC) via the m$^6$A mechanism. Specifically, CSC-induced hypomethylation in the promoter of *METTL3* causes its overexpression, which significantly increases m$^6$A formation in pri-miR-25. An RNA-binding protein NKAP acts as the m$^6$A reader and preferentially binds to the consensus motif RGm$^6$AC at the pri-miR-25 splicing site. NKAP-binding facilitates the interaction of pri-miR-25 with the miRNA microprocessor complex protein DGCR8 and promotes the maturation of miR-25-3p. Excessive miR-25-3p targets *PHLPP2* mRNA and substantially suppresses the expression of PH domain leucine-rich repeat protein phosphatase 2 (PHLPP2). Reduced PHLPP2 expression evokes the oncogenic AKT-p70S6K signaling. This METTL3-miR-25-3p-PHLPP2-AKT axis promotes the initiation and the progression of PDAC.

## Results

### CSC induces overexpression of oncogenic miR-25-3p in PDAC.
Microarray determination showed 26 miRNAs that had the expression levels significantly different in immortalized human pancreatic duct epithelial cells (HPDE6-C7) exposed to CSC compared with cells exposed to vehicle DMSO ($P < 0.05$, Student $t$ test). Among them, 16 were upregulated (fold change > 2) and 10 were downregulated (fold change < 0.5; Fig. 1a). These results were verified by the quantitative real-time PCR (qRT-PCR) analysis (Fig. 1b). The most significantly upregulated miRNA was miR-25-3p (fold change = 65, $P = 2.09 \times 10^{-6}$, Student $t$ test), with the induction being in a CSC dose-dependent manner

(Fig. 1c). Measurement of these 26 miRNAs in surgically removed non-tumor pancreatic tissue samples collected at the two cancer centers (total $N = 175$; Supplementary Table 1) showed that only the miR-25-3p level was significantly higher in smokers ($N = 67$) than in nonsmokers ($N = 108$; Fig. 1d) and the levels were also significantly higher in PDAC than in non-tumor samples in both patient groups (Fig. 1e). A significantly higher level of miR-25-3p in PDAC than in normal samples was seen in the public Gene Expression Omnibus (GEO) database (Supplementary Fig. 1a). Kaplan–Meier plots showed that patients with high miR-25-3p level (≥ median) in PDAC had shorter overall survival time than those with low level (< median) (Fig. 1f–h; Supplementary Table 2 and Supplementary Data 1). However, we did not find a significant association of the miR-25-3p levels with PDAC survival time in the TCGA data (Supplementary Fig. 1b), possibly due to the differences in other genetic or epigenetic alterations between different races of patients in study. Interestingly, we also found the existence of miR-25-3p in serum of healthy individuals (Supplementary Table 3) and the levels were strikingly higher in current smokers ($N = 22$) than in nonsmokers ($N = 23$) (Fig. 1i). We further investigated the effects of miR-25-3p on phenotypes of HPDE6-C7 and two PDAC cell lines PANC-1 and BXPC-3 and found that miR-25-3p overexpression substantially enhanced but knockdown significantly inhibited cell proliferation, migration, and invasion (Supplementary Fig. 2a–c). These effects were also seen in mice xenograft growth and metastasis models of PDAC cells with miR-25-3p overexpression or knockdown (Supplementary Fig. 2d, e).

### CSC promotes METTL3-mediated pri-miR-25 maturation.
To elucidate the mechanism for CSC in inducing miR-25-3p, we first analyzed the pri-miR-25, pre-miR-25, and miR-25-3p levels in HPDE6-C7 and pancreatic cancer cells exposed to CSC. The pri-miR-25 level was significantly decreased while the pre-miR-25 and miR-25-3p levels were prominently increased in cells with CSC exposure compared with cells without CSC exposure (Fig. 2a; Supplementary Fig. 3a), indicating that the miR-25-3p induction by CSC may be through aberrant pri-miR-25 processing and maturation. Since METTL3 plays a key role in miRNA maturation by catalyzing m$^6$A formation in RNAs, we examined whether METTL3 affects the CSC-induced miR-25-3p maturation. As a result, we found a coincidence of CSC-induced overexpression of miR-25-3p and *METTL3* but not *METTL14* and *WTAP* (Fig. 2b; Supplementary Fig. 3b). Overexpression of METTL3 in cells significantly decreased the pri-miR-25 level but increased both pre-miR-25 and miR-25-3p levels; while knocking down of METTL3 remarkably reduced both pre-miR-25 and miR-25-3p levels but increased the pri-miR-25 level (Supplementary Fig. 4a; Fig. 2c). Knocking down of *METTL3* also significantly inhibited CSC-induced miR-25-3p overexpression in HPDE6-C7 and pancreatic cancer cells (Fig. 2d). These results suggest that METTL3 may play an important role in the CSC-induced miR-25-3p overexpression.

### CSC-caused promoter hypomethylation upregulates *METTL3* expression.
We next wanted to explore how CSC causes METTL3 overexpression. In silico analysis proposed a CpG island ~0.2–0.6 kb upstream of the *METTL3* locus (Supplementary Fig. 5a), implying an epigenetic mechanism in its expression regulation. The sodium bisulfite sequencing showed that cells exposed to CSC had significantly lower methylation within this CpG island compared with cells without the exposure (Fig. 2e) and these results were all verified by the methylation-specific PCR analysis (Fig. 2e). In addition, the methylation levels within the *METTL3* CpG island in non-tumor pancreatic tissues were significantly

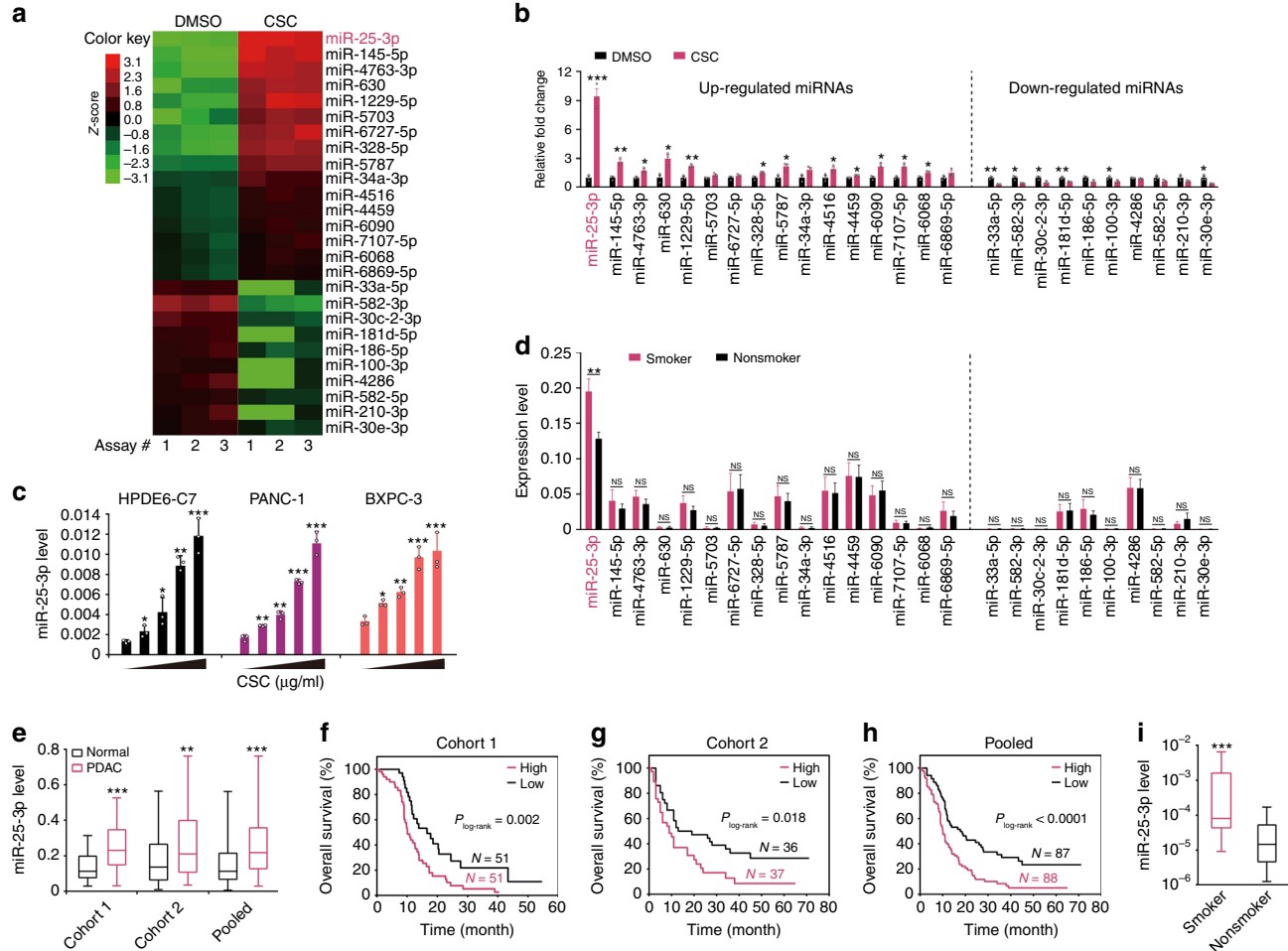

**Fig. 1** MiR-25-3p induction by CSC and its effect on survival time of individuals with PDAC. **a** Heatmap of significantly different expression of microRNAs (*P* < 0.05, fold change > 2 or < 0.5) detected with microRNA array in HPDE6-C7 cells exposed to CSC (100 μg/ml) or equal amount of DMSO as solvent control. The red (higher expression) or green (lower expression) color represents the normalized expression value of indicated microRNAs. **b** Expression levels of indicated miRNAs in HPDE6-C7 cell exposed to CSC (100 μg/ml) or equal amount of DMSO as vehicle control. Results represent means ± standard derivation (S.D.) from three independent measurements. **c** The induction of miR-25-3p overexpression in HPDE6-C7, PANC-1, and BXPC-3 cells was in a CSC dose (0, 0.1, 1.0, 10, and 100 μg/ml) dependent manner. Data represent mean ± S.D. from three independent experiments. **d** Expression levels of indicated miRNAs in non-tumor tissue samples of smokers (*N* = 67) or nonsmokers (*N* = 108). Results represent means ± S.D. from three independent measurements. **e** Aberrant overexpression of miR-25-3p in surgically removed PDAC samples compared with their paired non-tumor tissue samples in two patient groups and combined sample. **f–h** Kaplan–Meier estimates of survival time in two groups of patients with PDAC and combined sample by different miR-25-3p levels in tumor. The median survival time for individuals with high miR-25-3p levels (≥median) in Guangzhou, Beijing and combined sample was 10.1, 9.0, and 10.1 months compared with 16.6, 16.0, and 17.8 months in those with low miR-25-3p levels (<median), with the hazard ratio and 95% confidence interval (CI) of 2.20 (95% CI, 1.53–3.63), 1.82 (95% CI, 1.15–3.36), and 2.07 (95% CI, 1.50–3.09). **i** Significant difference in serum miR-25-3p levels among smokers (*N* = 22) and nonsmoker (*N* = 23). Data of (**e**) and (**i**) are displayed in min to max boxplot. The line in the middle of the box is plotted at the median while the upper and lower hinges indicated 25th and 75th percentiles. Student *t* tests were used in (**a**), (**b**), and (**c**) (NS, non-significant, *P < 0.05, **P < 0.01, and ***P < 0.001) and Wilcoxon rank-sum tests were used in (**d**), (**e**), and (**i**) (NS, non-significant, **P < 0.01, and ***P < 0.001)

lower in smokers than in nonsmokers (Fig. 2f). Quantitative chromatin immunoprecipitation (ChIP) assays showed that CSC substantially reduced the bindings of DNA methyltransferase 1 (DNMT1) and DNMT3a to the *METTL3* promoter (Fig. 2g). The reduced bindings were also found in pancreatic tissues of smokers compared with nonsmokers (Fig. 2h). We then sought to identify transcription factor(s) involved in *METTL3* overexpression and found that based on the suggestion of in silico analysis (Supplementary Fig. 5a), only *NFIC*, among other four potential transcription factors, was significantly and positively correlated with *METTL3* at the RNA level in both tumor and non-tumor pancreatic tissues, and this result was seen in TCGA and Genotype-Tissue Expression Project data (Supplementary

Fig. 5b). Knocking down of NFIC expression in HPDE6-C7 and PDAC cells resulted in substantially decreased METTL3 levels (Fig. 2i) and the transcriptional activity of *METTL3* promoter (Supplementary Fig. 5c). CSC exposure significantly increased NFIC enrichment within the *METTL3* regulatory region (Fig. 2j) and the transcriptional activity of *METTL3* promoter (Supplementary Fig. 5d), while the NFIC expression level was not changed (Supplementary Fig. 5e). Similar to the effect of *METTL3* depression, *NFIC* knockdown also substantially inhibited CSC-induced miR-25-3p overexpression in HPDE6-C7 and pancreatic cancer cells (Supplementary Fig. 5f). Remarkably, we detected a significantly more enrichment of NFIC within *METTL3* regulatory region in non-tumor pancreatic tissues from smokers

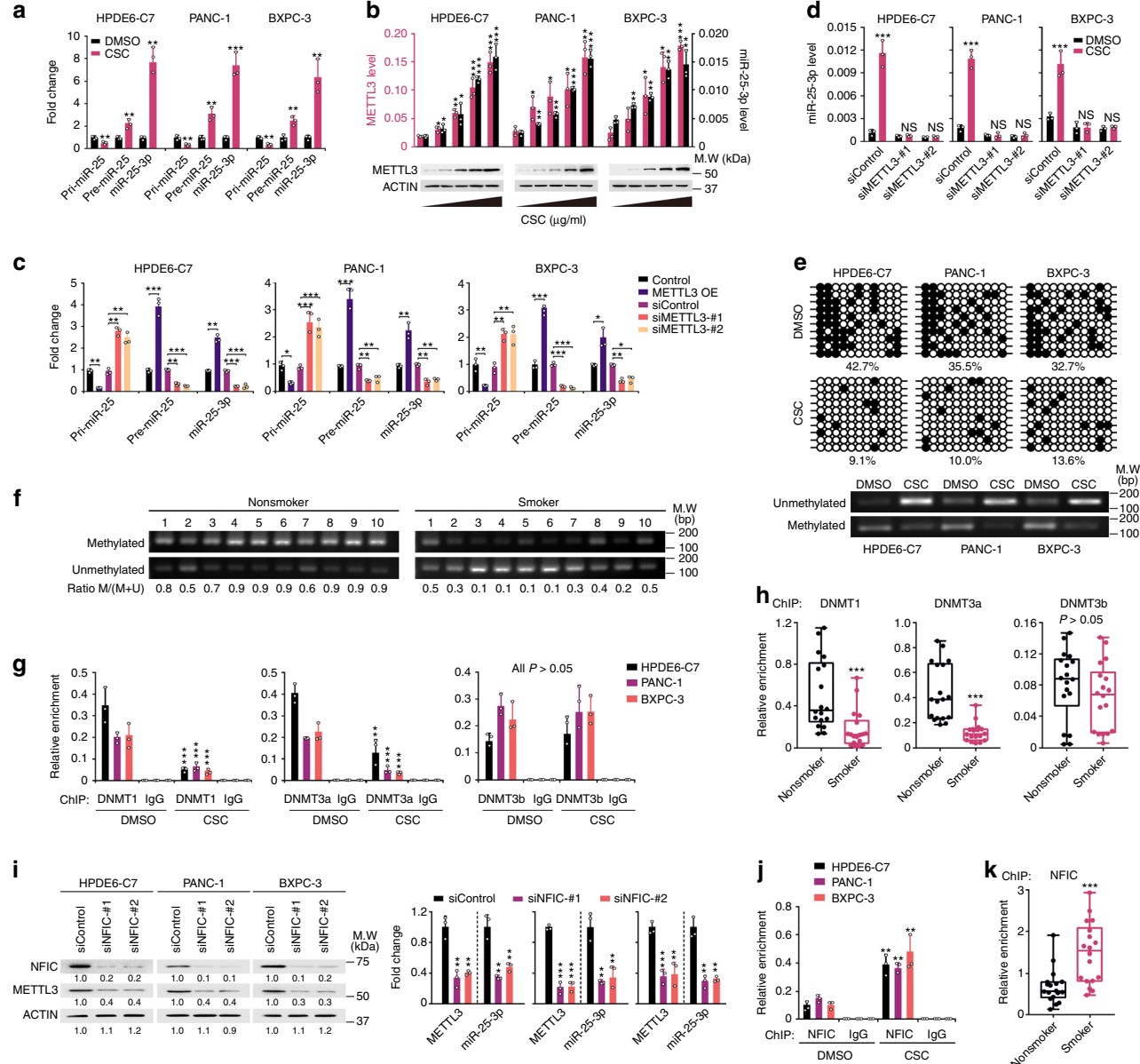

**Fig. 2** CSC promotes pri-miR-25 maturation by upregulating METTL3 expression. **a** Relative expression levels of pri-miR-25, pre-miR-25, and miR-25-3p in HPDE6-C7, PANC-1, and BXPC-3 cells exposed to CSC (100 μg/ml). **b** Effects of CSC on the expression of miR-25-3p and METTL3 mRNA (upper panel) and protein (lower panel) in HPDE6-C7, PANC-1, and BXPC-3 cells in a dose (0, 0.1, 1.0, 10, and 100 μg/ml) dependent manner. **c** METTL3 overexpression or knockdown significantly affected the expression of pri-miR-25, pre-miR-25, and miR-25-3p in PDAC cells. OE, overexpression. **d** METTL3 knockdown significantly affected the expression of miR-25-3p in HPDE6-C7, PANC-1, and BXPC-3 cells exposed to CSC. **e** CSC (100 μg/ml) induced hypomethylation of CpG island in the *METTL3* promoter region (between –284 and –564 bp from transcription start site) detected by bisulfate sequencing (upper panel) and methylation-specific PCR (lower panel) in HPDE6-C7 and PDAC cells. **f** Representative methylation-specific PCR (MSP) of non-tumor tissues from smokers or nonsmokers showing the *METTL3* CpG islands hypomethylation in smokers. Ratio (M/M + U) represents the methylated rate in indicated samples. **g** ChIP-qPCR assays showing decreased bindings of DNMT1 and DNMT3a but not DNMT3b within the *METTL3* CpG islands caused by CSC. **h** Quantitative ChIP analysis of DNMT1, DNMT3a, and DNMT3b levels within the *METTL3* regulatory region in non-tumor tissues from 36 patients (18 smokers and 18 nonsmokers). **i** NFIC knockdown significantly affected the expression levels of METTL3 proteins (left panel) and RNA and miR-25-3p (right panel). Results of RNA represent means ± S.D. from three independent experiments. **j** ChIP-qPCR assays showing significantly increased direct binding of NFIC to *METTL3* promoter in HPDE6-C7 and PDAC cells exposed to CSC. **k** Quantitative ChIP analysis showing significant difference in the NFIC levels within the *METTL3* regulatory region between smokers' and nonsmokers' non-tumor pancreatic tissues (both $N = 18$). Data of (**h**) and (**k**) are displayed in min to max boxplot. The line in the middle of the box is plotted at the median while the upper and lower hinges indicated 25th and 75th percentiles. Values are the mean ± S.D. from three independent experiments, and all statistic analyses in this figure are Student *t* test. *$P < 0.05$, **$P < 0.01$, ***$P < 0.001$ and NS, not significant as compared with the corresponding control

than that from nonsmokers (Fig. 2k). Together, these results demonstrate that CSC causes *METTL3* hypomethylation, which facilitates the recruitment of transcription factor NFIC to induce *METTL3* overexpression.

**Modification of m⁶A mediates METTL3-caused pri-miR-25 maturation**. We then explored whether m⁶A affects miR-25-3p maturation. We first sought for the m⁶A site on pri-miR-25 by using the m⁶A individual-nucleotide resolution cross-linking and immunoprecipitation sequencing (miCLIP-seq) approach[27]. The results showed an m⁶A exactly localized at the pri-miR-25 splicing site (Fig. 3a). Subsequent m⁶A-specific RNA immunoprecipitation (RIP) coupled qRT-PCR analysis showed that the pri-miR-25 m⁶A levels in PDAC samples were significantly higher than that in paired non-tumor samples (Fig. 3b). In vitro cell models examining the effect of METTL3-mediated m⁶A

formation on pri-miR-25 processing showed that the [m⁶A]pri-miR-25 levels were significantly increased in cells with CSC exposure compared with cells without CSC exposure (Fig. 3c). Furthermore, METTL3 overexpression significantly increased but knockdown remarkably reduced [m⁶A]pri-miR-25 levels in both cell lines (Fig. 3d). These results suggest a vital role of METTL3 in m⁶A formation and miR-25-3p maturation. To verify a direct role of m⁶A in miR-25-3p maturation, we performed in vitro RNA processing assays using in vitro transcribed pri-miR-25 with or without m⁶A and whole cell lysates of 293T cells overexpressing DGCR8 and DROSHA. [m⁶A]pri-miR-25 was more rapidly and efficiently processed to pre-miR-25 and miR-25-3p compared with its unmethylated counterpart (Fig. 3e, f). When the potential METTL3-catalyzing motif in pri-miR-25 was mutated, the processing rate of pri-miR-25 to its mature forms was significantly decreased (Fig. 3g, h). These in vitro results were in line with the

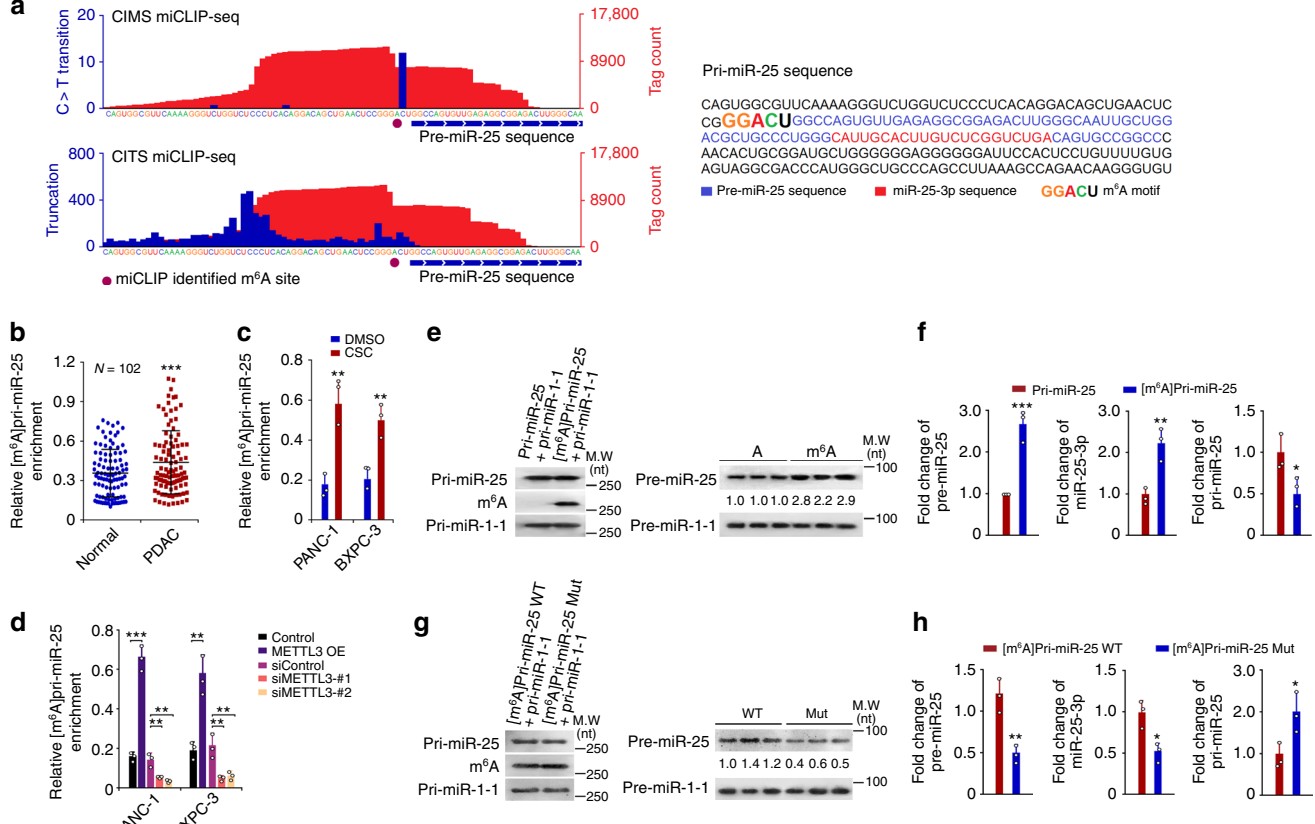

**Fig. 3** m⁶A modification is a major mechanism for METTL3-promoted pri-miR-25 maturation. **a** Left: Identification of m⁶A by m⁶A individual-nucleotide resolution cross-linking and immunoprecipitation sequencing (miCLIP-seq). The m⁶A residues were detected by cross-linking induced mutation sites (CIMS) and cross-linking induced truncation sites (CITS) in pri-miR-25. Red tracks of miCLIP-seq are unique tag coverage and blue tracks represent C>T transition or truncation, respectively. Filled purple circles denote miCLIP-called m⁶A and the horizontal blue bars indicate transcript models of pre-miR-25. Right: The sequences of pre-miR-25 and miR-25-3p are highlighted, respectively, by different colors, and the m⁶A motif (GGACU) is located at the putative splicing site. **b** PDAC had significantly higher levels of [m⁶A]pri-miR-25 than paired non-tumor tissues. [m⁶A]pri-miR-25 was detected by immunoprecipitation followed by qRT-PCR analysis. ***P < 0.001 by Wilcoxon rank-sum test. **c**, **d** Effects of CSC exposure (100 μg/ml) and overexpression or knockdown of METTL3 on the levels of [m⁶A]pri-miR-25 in PDAC cells. Data represent mean ± S.D. from three independent experiments. **e**, **f** m⁶A modification substantially enhanced pri-miR-25 processing and maturation in an in vitro reaction system containing starting materials of pri-miR-25 or [m⁶A]pri-miR-25 and the whole cellular lysates of 293T cells transfected with plasmids carrying DROSHA and DGCR8. Pri-miR-1-1 which has no methylation site was included as a control. **e** Northern blot detection of starting materials (left panel) and the levels of resultants pre-miR-25 and pre-miR-1-1 in the reaction. **f** Quantification of pre-miR-25, miR-25-3p, and pri-miR-25 in the reaction mixture. Data represent fold change ± S.D. relative to pri-miR-25 as starting material. **g**, **h** Mutation of [m⁶A]pri-miR-25 at METTL3-recognizing site abolished pri-miR-25 processing and maturation in the in vitro reaction system. **g** Northern blot detection of starting materials (left panel) and the levels of resultants pre-miR-25 and pre-miR-1-1 in the reaction. **h** Quantification of pre-miR-25, miR-25-3p, and pri-miR-25 in the reaction mixture. Data represent fold change ± S.D. relative to [m⁶A]pri-miR-25 WT as starting material from three experiments. All statistic analyses in this figure are Student *t* test unless specified. *P < 0.05, **P < 0.01, and ***P < 0.001 compared with the corresponding control

findings in vivo and in cell models, demonstrating that higher miR-25-3p levels in smokers and in PDAC might be the consequence of enhanced pri-miR-25 processing via m6A stimulated by cigarette smoke-induced overexpressed METTL3.

**NKAP is a reader of m6A in pri-miR-25 maturation.** Although RNA-binding protein HNRNPA2B1 is an m6A reader that

facilitates processing of a subset of miRNAs whose maturation is dependent on METLL3-mediated m6A modification[24], the m6A readers for many other pri-miRNAs including pri-miR-25 are still undiscovered. By mass spectrometry analyses of proteins generated by RNA pulldown using a 50-bp pri-miR-25 or [m6A]pri-miR-25 and co-IP with anti-DGCR8 antibody and subsequent integrative analysis (Fig. 4a; Supplementary Data 2—4), we

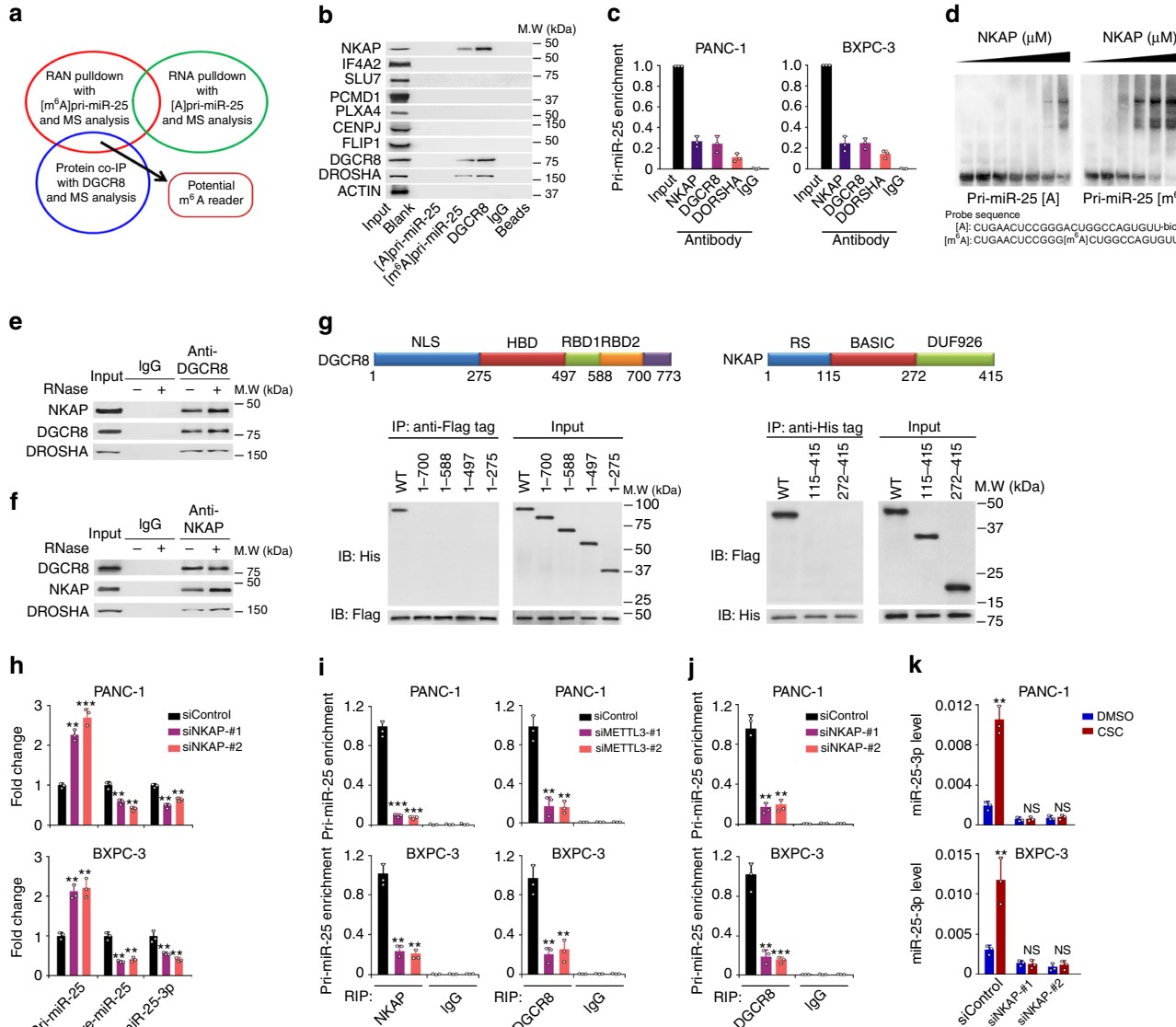

**Fig. 4** Identification of NKAP as a mediator of m6A in pri-miR-25 maturation. **a** Schematic of RNA pulldown and immunoprecipitation experiment and proteomic screening by mass spectrometry (MS) for the identification of [m6A]pri-miR-25 reader. **b** Western blot analysis of potential m6A readers obtained from proteomic screening as indicated in (**a**) shows specific association of NKAP with both [m6A]pri-miR-25 and DGCR8. **c** RNA immunoprecipitation assays with m6A antibody show association of NKAP (FLAG-tagged) with pri-miR-25 in PDAC cells. Data from three independent experiments represent RNA levels associated with NKAP relative to input. Antibody against DGCR8 or DROSHA was included as positive control and IgG served as negative control. **d** Electrophoretic mobility shift assays of recombinant NKAP with unmethylated or methylated pri-miR-25 probes. The probes were maintained constantly while a gradient of 0–10 μM recombinant NKAP was added to the reactions. **e**, **f** Reciprocal immunoprecipitation assays show interaction among DGCR8, DROSHA, and NKAP (FLAG-tagged) in PANC-1 cells in the presence of RNase A. **g** Confirmation of binding of DGCR8 to NKAP by truncation mapping test. Upper panel shown are the schematic of the domain structures of DGCR8 and NKAP protein. Lower panel: Western blot analysis of constructs for His-tagged DGCR8 (wild-type versus domain truncation mutants) and FLAG-tagged NKAP or for FLAG-tagged NKAP (wild-type versus domain truncation mutants) and His-tagged DGCR8 that were co-transfected into PANC-1 cells, respectively. **h** Knockdown of NKAP expression in PDAC cells substantially decreased pre-miR-25 and miR-25-3p levels but substantially increased the level of pri-miR-25. **i** RNA immunoprecipitation assays show significantly decreased enrichment of pri-miR-25 with NKAP (left panel) or DGCR8 (right panel) in PDAC cells with METTL3 knockdown. **j** RNA immunoprecipitation assays show significantly decreased enrichment of pri-miR-25 with DGCR8 in PDAC cells with NKAP knockdown. **k** NKAP knockdown affected the expression of miR-25-3p in PDAC cells exposed to CSC. Values of (**h–k**) are the mean ± S.D. from three independent experiments. All statistic analyses in this figure are Student t test. *P < 0.05, **P < 0.01, and ***P < 0.001 compared with the corresponding control

identified 22 proteins including 9 nuclear proteins potentially interacting with DGCR8 and [m$^6$A]pri-miR-25 (Supplementary Table 4). Western blotting and RIP-coupled qRT-PCR analysis showed that among nine nuclear proteins, only NF-κB associated protein (NKAP) bound both [m$^6$A]pri-miR-25 and DGCR8 (Fig. 4b, c). RNA electrophoretic mobility shift assays verified that NKAP preferentially bound [m$^6$A]pri-miR-25 but not unmethylated pri-miR-25 (Fig. 4d). Co-IP assays showed a persisting interaction between NKAP and DGCR8 despite ribonuclease treatment (Fig. 4e, f), suggesting that this is a protein–protein association. Protein truncation mapping test confirmed this finding (Fig. 4g). Similar to METTL3 depletion, NKAP knocking down (Supplementary Fig. 4b) substantially increased the pri-miR-25 level but decreased the pre-miR-25 and miR-25-3p levels in cells (Fig. 4h). We also found that METTL3 or NKAP depletion substantially reduced the interaction of endogenous DGCR8 with pri-miR-25 (Fig. 4i, j), indicating that it was NKAP that recruits DGCR8 to interact with pri-miR-25. Moreover, NKAP knocking down remarkably inhibited CSC-induced miR-25-3p formation in PDAC cells (Fig. 4k).

We also performed NKAP-iCLIP to assess the abundance of the m$^6$A site in pri-miR-25 and found that NKAP sturdily binds the pri-miR-25 m$^6$A site (Fig. 5a). Analysis of the public DGCR8 HITS-CLIP data (GSE61979)[28] showed that the DGCR8-binding site on pri-miR-25 overlaps both NKAP-binding site and pri-miR-25 m$^6$A site (Fig. 5b). Cells exposed to CSC had significantly enhanced NKAP binding to the pri-miR-25 m$^6$A site (Fig. 5c). Similar results were seen in cells without CSC exposure but overexpressing METTL3 (Fig. 5d). Furthermore, we observed a substantial overlap of the m$^6$A sites obtained by miCLIP and the NKAP-binding sites obtained by iCLIP (Fig. 5e, f). Notably, a highly significant enrichment of RGAC motifs at NKAP-binding footprints (CIMS or CITS along with a 5-nucleotide flanking sequence) was identified by iCLIP-sequencing ($P < 10^{-12}$, Wilcoxon rank-sum test, Fig. 5g), suggesting that NKAP preferentially binds the m$^6$A core motif RGAC. Also, we observed a high binding intensity for NKAP centering at the m$^6$A residues and vice versa for m$^6$A (Fig. 5h, i) and a high binding intensity of NKAP was located around the splicing factors-binding sites (Fig. 5j), consistent with the NKAP role in RNA splicing. To examine whether the NKAP role in RNA splicing is dependent on m$^6$A modification, we profiled the NKAP-binding signal around splicing sites neighboring m$^6$A residues (in a 100-nucleotide flanking region of m$^6$A residues) and splicing sites not neighboring m$^6$A residues. Substantially higher NKAP-binding signal was detected in splicing sites near the m$^6$A residues

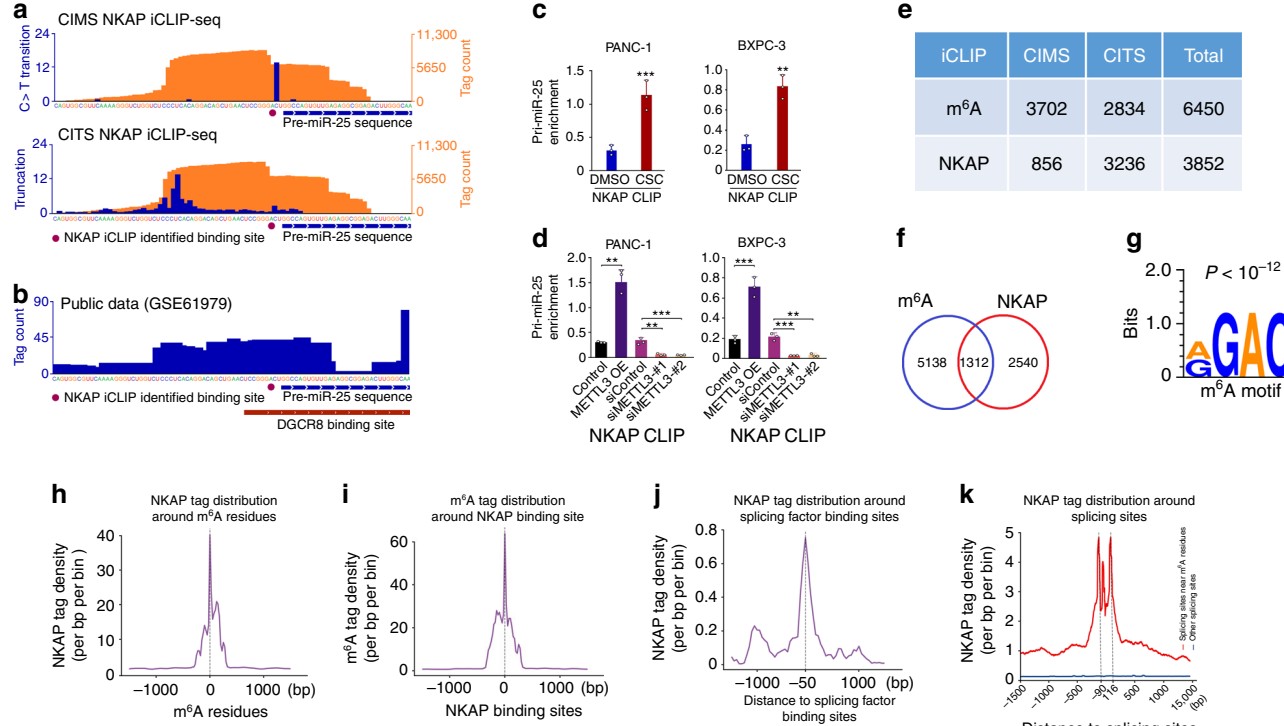

**Fig. 5** NKAP is a reader of m$^6$A on pri-miR-25. **a** The NKAP-binding sites on pri-miR-25 detected by CIMS and CITS miCLIP-seq. Orange tracks of NKAP-iCLIP-seq are unique tag coverage and blue tracks represent C>T transition and truncation, respectively. Filled purple circles denote iCLIP-called NKAP-binding site and the horizontal blue bars indicate the transcript model of pre-miR-25. **b** The DGCR8-binding sites on pri-miR-25 detected by DGCR8 HITS-CLIP. Filled purple circles denote iCLIP-called NKAP-binding site and the horizontal blue and red bars represent the transcript model of pre-miR-25 and the DGCR8-binding site, respectively. **c** In vivo binding of NKAP-FLAG to pri-miR-25 in PDAC cells exposed to CSC or DMSO detected by CLIP-qPCR. Values are the mean ± S.D. from three independent experiments. **d** The levels of NKAP-FLAG bound to pri-miR-25 were significantly elevated by overexpression of *METTL3*, but significantly decreased by knockdown of *METTL3* in both PANC-1 (left panel) and BXPC-3 (right panel) cells. Values are the mean ± S.D. from three independent experiments. **e** Data of m$^6$A residues and NKAP-binding sites identified by the analysis of cross-linking-induced mutation sites (CIMS) or cross-linking-induced truncation sites (CITS) in iCLIP-seq data. **f** Venn diagram showing overlap between m$^6$A residues and NKAP-binding sites. **g** Significant enrichment of RGAC motif is in NKAP footprints (CIMS or CITS along with 5-nucleotide flanking sequences) analyzed by MEME SUIT with default parameters. **h** Shown is the intensity of NKAP binding centered at m$^6$A residues. **i** Shown is the intensity of m$^6$A CLIP signal centered at NKAP-binding sites. **j** Shown is the distribution of NKAP iCLIP tag density around the splicing factor binding sites. **k** Shown is the binding intensity of NKAP at splicing sites near or not near m$^6$A residues. All statistic examinations in this figure are Student *t* test unless specific. *$P < 0.05$, **$P < 0.01$, and ***$P < 0.001$ compared with the corresponding control

compared with those far from the m[6]A residues, indicating that the NKAP role in RNA splicing is m[6]A-dependent (Fig. 5k). Taken together, these results strongly support that NKAP acts as both m[6]A reader and RNA splicing factor.

**MiR-25-3p targets *PHLPP2* and evokes AKT-p70S6K signaling.** We applied four publicly available algorithms to call the potential miR-25-3p target genes and the results (Supplementary Fig. 6a) suggest that PH domain leucine-rich-repeats protein phosphatase 2 (PHLPP2), a tumor suppressor that inhibits cancer cell proliferation and invasion[29,30] may be the target. We thus performed reporter gene assays with the vectors containing the *PHLPP2* 3′-UTR fragments (Supplementary Fig. 6b). A significant and dose-dependent reduction in reporter activity for vector containing the *PHLPP2* 3′-UTR compared with control in the presence of miR-25-3p in PDAC cells was observed and this reduction effect could be rescued by the addition of miR-25-3p inhibitor (Fig. 6a, b). When the miR-25-3p-binding sites in the 3′-UTR were mutated (Supplementary Fig. 6c), a dramatic relief of reporter gene silencing was achieved (Fig. 6c), suggesting that these are probably the true miR-25-3p targeting sites. Further experiments showed that both PHLPP2 mRNA and protein levels were remarkably declined in cells with miR-25-3p overexpression but dramatically increased in cells with miR-25-3p knockdown (Fig. 6d). The analysis of RIP with Ago2 antibody followed by qRT-PCR revealed a remarkable increase in recruitment of *PHLPP2* mRNA to the miRNA complex in cells overexpressing miR-25-3p (Fig. 6e).

Since AKT signaling is regulated by PHLPP, we examined whether PHLPP2 affects the AKT-p70S6K signaling and whether miR-25-3p affects the PHLPP2 function in PDAC cells. We found that knockdown of PHLPP2 expression by siRNA substantially activated the signaling as indicated by increased AKT and p70S6K phosphorylation (Fig. 6f). The effect of miR-25-3p on AKT-p70S6K signaling was evident in cells with miR-25-3p knockdown or overexpression (Fig. 6g). Rescue assays showed that knockdown of PHLPP2 partially reversed the effects of miR-25-3p knockdown on the suppression of AKT-p70S6K signaling and the malignant phenotypes of PDAC cells (Figs. 6h–j). These results suggest miR-25-3p as an upstream activator of AKT signaling acting through inhibiting PHLPP2 production. We further found that the AKT inhibitors MK-2206 and GDC-0068 could substantially abolish the effects of miR-25-3p on the AKT-p70S6K signaling and the malignant phenotypes of PDAC cells (Supplementary Fig. 7a–c). In addition, cells treated with the p70S6K inhibitor rapamycin substantially abolished the miR-25-3p-enhanced p70S6K activation (Supplementary Fig. 7d) and the malignant phenotypes (Supplementary Fig. 7b, c).

Importantly, we found that cells exposed to CSC had substantially elevated levels of phosphorylated AKT and p70S6K, which were concordant with overexpressed miR-25-3p and depressed PHLPP2 (Fig. 7a). In line with these molecular results, CSC exposure significantly promoted the malignant phenotypes of PDAC cells (Fig. 7b, c). Together, these results indicate that CSC-induced excessive maturation of miR-25-3p evokes oncogenic AKT-p70S6K signaling via targeting PHLPP2. We further examined the relationships among miR-25-3p, METTL3, and PHLPP2 levels in clinical tissue samples. A positive correlation between miR-25-3p and *METTL3* RNA levels and a negative correlation between *PHLPP2* RNA levels and miR-25-3p or *METTL3* RNA levels were detected in both PDAC and non-tumor pancreatic tissues from smokers (Fig. 7d) and nonsmokers (Fig. 7e). We also detected METTL3, PHLPP2, p-AKT, and p-p70S6K protein levels in non-tumor pancreatic tissue samples from smokers and nonsmokers and found that all of

them were significantly higher in smokers than in nonsmokers, except for PHLPP2 that was lower in smokers compared with nonsmokers (Fig. 7f, g). In addition, a positive correlation between METTL3 and p-AKT or p-p70S6K levels and a negative correlation between PHLPP2 and METTL3 or p-AKT or p-p70S6K levels were observed in non-tumor pancreatic tissues from smokers (Fig. 7h) and nonsmokers (Fig. 7i). These results suggest that the vicious METTL3-miR-25-3p-PHLPP2-AKT regulatory axis is probably present in vivo in pancreatic tissues.

## Discussion
In this study, we identified a panel of aberrantly expressed miRNAs in pancreatic duct epithelial cells exposed to CSC and the most significant one is miR-25-3p, an oncogenic miRNA reported in many types of human cancer[31–36]. We found that the miR-25-3p level is significantly higher in smokers than in non-smokers and in PDAC than in non-tumor tissues and the elevated level is correlated with shorter survival time of patients. Forced expression of miR-25-3p promoted PDAC cell proliferation and metastasis in vitro and in vivo in mice. The present study has several novel discoveries. First, we found that CSC induces DNA hypomethylation in the promoter of *METTL3*, resulting in aberrant overexpression of METTL3 enzyme, which substantially promotes miR-25-3p maturation from its primary transcript via enhanced m[6]A modification. Second, we identified NKAP as an m[6]A reader for m[6]A-mediated pri-miR-25 processing. Third, we identified *PHLPP2* as a target of miR-25-3p. One of the oncogenic effects of aberrant expression of miR-25-3p is to inhibit *PHLPP2* and activate the AKT-p70S6K oncogenic signaling, forming a vicious METTL3-miR-25-3p-PHLPP2-AKT axis to promote PDAC development in smokers (Fig. 7j).

Although cigarette smoking is an established etiological factor for many types of human cancer including PDAC and studies have shown that PDAC patients who smoked have worse prognosis[37,38], the mechanisms for the effects of cigarette smoking remain to be elucidated. Since cigarette smoking does not cause PDAC through mutating the known driver genes such as *TP53* and *KRAS*[11,12], epigenetic effects might play a role in the development of smoking-related PDAC. The epigenetic effects of gene expression resulting from cigarette smoking have been previously reported, but the major finding was how smoking affects the expression of DNA methylation related-proteins or the recruitment of these proteins to the CpG islands. In the present study, we found that CSC is able to epigenetically induce aberrant overexpression of METTL3, a methyltransferase for m[6]A modification in RNAs. The overexpression of METTL3 enhances m[6]A modification in pri-miR-25 and accelerates its maturation to miR-25-3p. Furthermore, we found that smokers had remarkably higher serum miR-25-3p levels than nonsmokers. These results strongly suggest an important epigenetic mechanism of cigarette smoking in the induction of oncogenic miR-25-3p. Since smoking continues as a risk factor for PDAC even after 5 years or more of cigarette cessation[9,10] and the disrupted expression of some miRNAs may remain in former smokers[39] it would be interesting to investigate whether the high plasma miR-25-3p level retains in former smokers who quit smoking more than 5 years ago and whether it could serve as a biomarker for risk assessment.

The modification of m[6]A in RNAs is a common epigenetic regulation involved in a variety of cellular processes[21,22], such as RNA stability[40–42], splicing[43], and protein production[42,44,45], all of which, if aberrant, may cause diseases including malignancies[46–49]. In the present study, we have linked aberrant m[6]A modifications in pri-miR-25 to the development and progression of PDAC via AKT signaling pathway, which plays a crucial role in multiple types of malignances[50]. Aberrant activation of the AKT

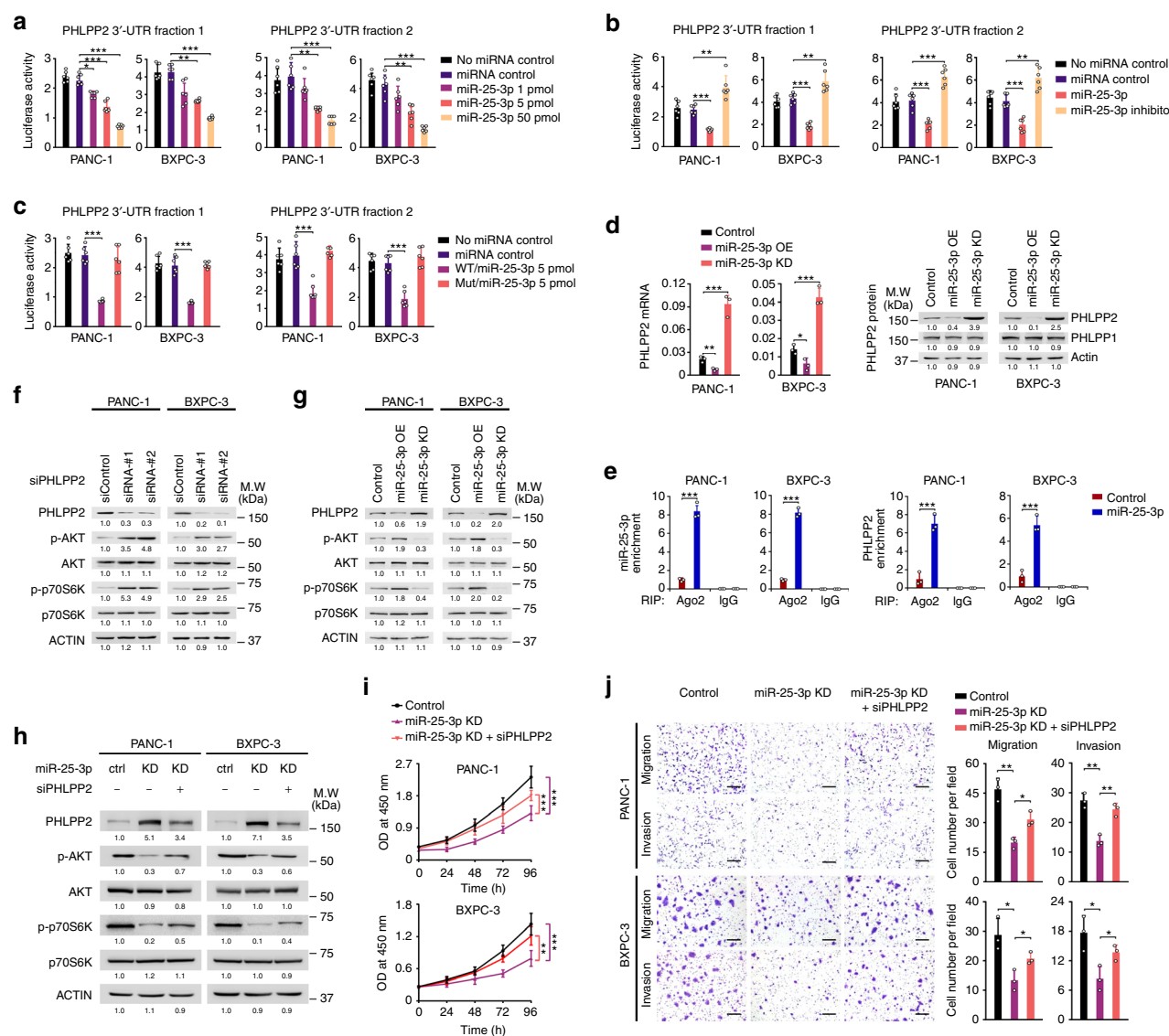

**Fig. 6** MiR-25-3p targets *PHLPP2* and evokes oncogenic AKT-p70S6K signaling. **a** Relative reporter gene activity of psiCHECK2 vector bearing *PHLPP2* 3′-UTR fraction 1 (left panel) or fraction 2 (right panel) in PDAC cells co-transfected with increasing amounts (1, 5, and 10 pmol) of miR-25-3p mimic. **b** Relative reporter gene activity of constructs bearing *PHLPP2* 3′-UTR fraction 1 (left panel) or fraction 2 (right panel) in PDAC cells co-transfected with 5 pmol of miR-25-3p mimic or its inhibitor. **c** Reporter gene activity of psiCHECK2 vector bearing *PHLPP2* 3′-UTR fraction 1 (left panel) or fraction 2 (right panel) or their mutant counterparts in PDAC cells in the presence of 5 pmol miR-25-3p mimic. Results in (**a**), (**b**), and (**c**) represent means ± S.D. from three experiments and each had six replicates. **d** Stable overexpression or knockdown of miR-25-3p on the levels of PHLPP2 RNA (left panel) and protein (right panel). Results of mRNA represent means ± S.D. from three independent experiments. **e** RNA immunoprecipitation assays show significantly increased association with Ago2 of both miR-25-3p (left panel) and *PHLPP2* mRNA (right panel) in PDAC cells overexpressing miR-25-3p. Results represent means ± S.D. from three independent experiments. **f** MiR-25-3p evokes AKT-p70S6K signaling via PHLPP2. Knockdown of PHLPP2 substantially enhanced AKT and p70S6K phosphorylation activation in PDAC cells. **g** Overexpression or knockdown of miR-25-3p had substantial effects on PHLPP2 expression and phosphorylation of its downstream signaling modules. **h** Knockdown of PHLPP2 expression substantially rescued the activation of AKT and p70S6K suppressed by miR-25-3p knockdown in PDAC cells. Ctrl control, KD knockdown. **i** Knockdown of PHLPP2 expression significantly reversed the effects of miR-25-3p knockdown on PDAC cell proliferation. Results represent mean ± S.D. from three independent experiments. **j** Knockdown of PHLPP2 expression significantly reversed the effects of miR-25-3p knockdown on PDAC cell migration and invasion. Left panel shows representative images of transwell assays and right panel shows quantitative statistics. Results are mean ± S.D. from three random fields. All statistic analyses in this figure are Student *t* test. *$P < 0.05$, **$P < 0.01$, and ***$P < 0.001$ compared with the corresponding control

pathway was detected in up to 70% of PDAC and the level of phosphorylated AKT in PDAC is an independent contributor to poor prognosis of pancreatic cancer[51]. However, how this oncogenic signaling pathway is activated in PDAC and by what is largely unknown. The results of the present study have demonstrated that aberrant overproduction of miR-25-3p induced by cigarette smoke can trigger the AKT-p70S6K pathway via

suppressing PHLPP2, leading to the formation of METTL3-miR-25-3p-PHLPP2-AKT axis, which is probably one of the underlying molecular mechanisms by which smoking induces PDAC tumorigenesis and progression.

Another interesting finding in the present study is a new m⁶A reader. Previous studies have reported several nuclear proteins that may serve as m⁶A readers including YTH family

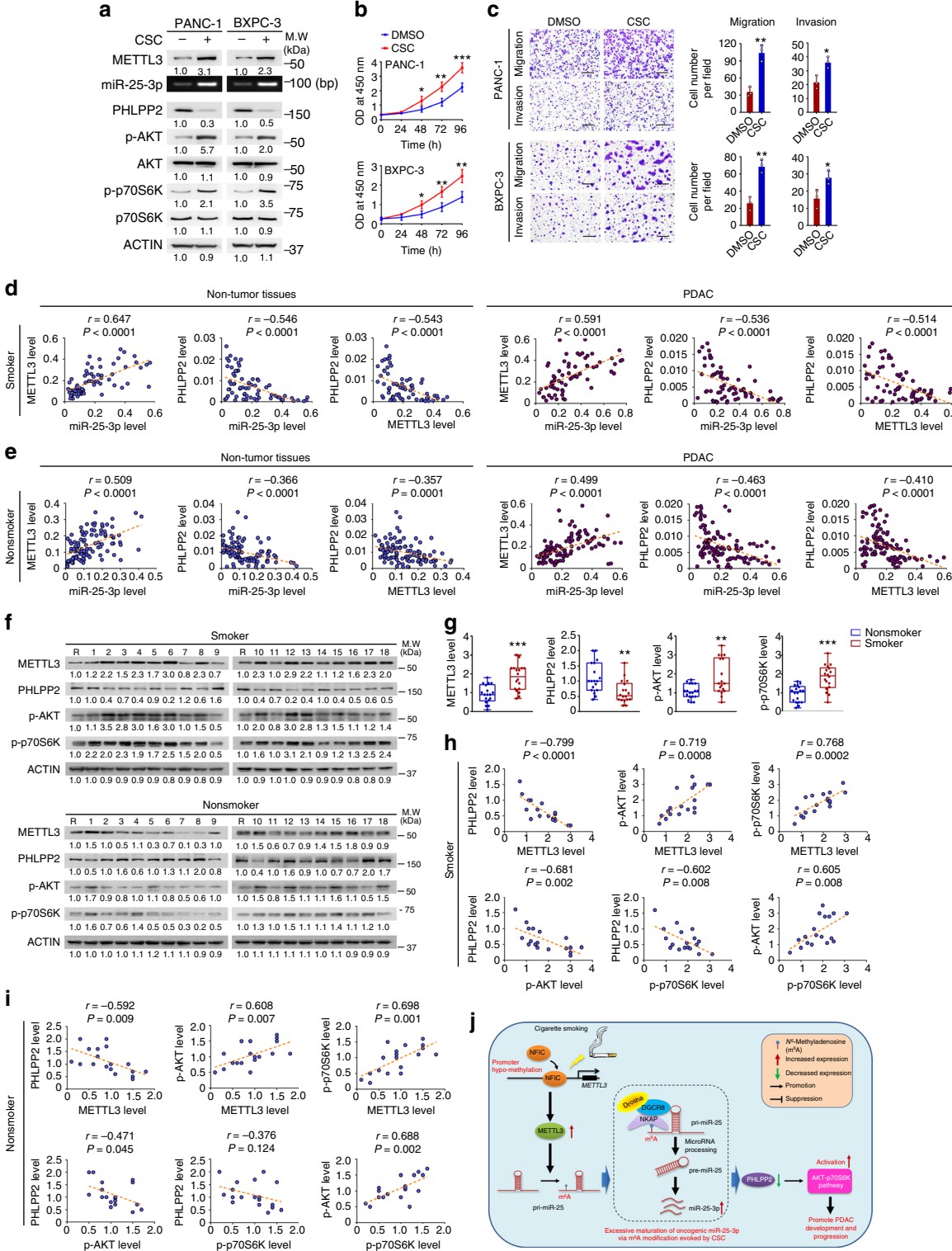

proteins[40,43,44] and, more recently, HNRNPA2B1 protein[24]. Under our experimental conditions, we found that NKAP preferentially interacts with both [m6A]pri-miR-25 and DGCR8, promoting the maturation process of pri-miR-25. NKAP was initially identified as an activator of NF-κB and the Notch pathway[52,53] and its ability to interact with RNAs and RNA-binding proteins has recently been discovered. For instance, NKAP can bind pre-mRNA and splice mRNA by recruiting, activating, or stabilizing the RNA processing factors. The RS and DUF926 domains in NKAP act as RNA-binding domains and the former also mediates NKAP to interact with other proteins[54]. By showing direct binding of NKAP to the consensus m6A motif RGm6AC, we have provided evidence that NKAP is likely m6A reader. This notion is further confirmed by miCLIP- and NKAP iCLIP-sequencing, which showed in vivo binding of NKAP to the m6A site in pri-miR-25 at the single-nucleotide level. We have also demonstrated that NKAP physically interacts with DGCR8. Together, these results provide strong evidence that nuclear protein NKAP is a likely reader of m6A in pri-miR-25. Although the exact molecular mechanism for the association of

**Fig. 7** Cigarette smoking evokes the activation of AKT-p70S6K signaling via miR-25-3p in PDAC. **a** CSC evokes the activation of AKT-p70S6K signaling via miR-25-3p in PDAC cells. **b**, **c** Exposure to CSC significantly enhanced the ability of proliferation (**b**) and migration and invasion (**c**) in PDAC cells. Scale bars, 200 μm. Right panel shows quantitative statistics. Results are mean ± S.D. from three random fields. **d** Correlations between miR-25-3p and *METTL3* RNA levels and between *PHLPP2* RNA and miR-25-3p or *METTL3* RNA levels in smokers' non-tumor pancreatic tissues ($N = 67$, left panel) and PDAC ($N = 67$, right panel) by Pearson's test. **e** Correlations between miR-25-3p and *METTL3* RNA levels and between *PHLPP2* RNA and miR-25-3p or *METTL3* RNA levels in nonsmokers' non-tumor pancreatic tissues ($N = 108$, left panel) and PDAC ($N = 108$, right panel) by Pearson's test. **f** Western blot analysis of protein expression levels of indicated genes in non-tumor tissues from nonsmokers ($N = 18$, upper panel) or smokers ($N = 18$, lower panel). R, the same positive reference sample for loading adjustment on each gel. Each protein band was semi-quantified by gray density and the value for each band is relative to density of both β-ACTIN and the corresponding band of R. **g** Significant difference in the expression levels of indicated proteins in non-tumor tissues from nonsmokers or smokers (both $N = 18$). Data are displayed in min to max boxplot. The line in the middle of the box is plotted at the median while the upper and lower hinges indicated 25th and 75th percentiles. *$P < 0.05$, **$P < 0.01$, and ***$P < 0.001$ by Student $t$ test. **h** Significant Pearson correlations among the expression levels of METTL3, PHLPP2, p-AKT, and p-p70S6K proteins in smokers' non-tumor tissues ($N = 18$). **i** Shown are Pearson correlations between METTL3, PHLPP2, p-AKT, and p-p70S6K protein expression levels in non-tumor tissue specimens from nonsmokers ($N = 18$). Data of (**g–i**) were from western blot analyses showing in (**f**). **j** A proposed action model for excessive miR-25-3p maturation via m6A modification stimulated by cigarette smoking in the development and progression of PDAC

[m6A]pri-miR-25, NKAP, and DGCR8 remains to be clarified, we speculate that NKAP recognizes the m6A mark and acts as a bridge connecting [m6A]pri-miR-25 with DGCR8 and thus recruits the microprocessor complex to perform maturation processing.

Since m6A modification also exists in many other primary miRNAs, it would be interesting and important to clarify whether NKAP can recognize m6A in primary miRNAs other than pri-miR-25 and therefore play a broader role in miRNA processing. Further studies on how this molecule may affect other miRNA processing would expand our knowledge about m6A modification in regulating miRNA biogenesis and the development of cancer. The present study only used PDAC as a disease model to examine the effect of aberrant m6A modification in miRNAs by CSC. Since cigarette smoke is an etiological factor for multiple types of human cancer and aberrant overexpression of miR-25-3p is often detected in tumors, it would be interesting to examine and compare if the epigenetic mechanism detected in PDAC in the present study also exists in other smoking-related cancers.

In conclusion, we have identified a vicious axis of METTL3-miR-25-3p-PHLPP2-AKT in cells exposed to cigarette smoke, which might be implicated in the development and progression of smoking-related PDAC. Our results also suggest that aberrant m6A modification in RNAs may play an important role in PDAC and other types of cancer attributed to cigarette smoking.

## Methods

**Collection of bio-samples and clinical data.** Surgically removed 175 PDAC and paired non-tumor tissue samples were obtained from individuals who underwent pancreatectomy at Sun Yat-sen Memorial Hospital, Sun Yat-sen University (Guangzhou, China, $N = 102$) and Cancer Hospital, Chinese Academy of Medical Sciences (Beijing, China, $N = 73$) between 2010 and 2016 (Supplementary Table 1). PDAC was diagnosed by the histopathological examination. Demography characteristics and clinical information of individuals were obtained from medical records. All the individuals underwent pancreatectomy received no chemotherapy or radiotherapy before surgery. The bio-specimens from each individual were collected at the time of surgery. Survival time of individuals with PDAC was measured from the date of diagnosis to the date of last follow-up or death (Supplementary Data 1). Whether and when a subject had died was obtained from inpatient and outpatient records, subject's family, or through follow-up telephone calls. For determination of miR-25-3p in serum, current smokers ($N = 22$) and nonsmokers ($N = 23$) were recruited from a general population (Supplementary Table 3). Informed consent was obtained from each participant, and this study was approved by the Institutional Review Board of the Sun Yat-sen Memorial Hospital and Chinese Academy of Medical Sciences, Cancer Hospital.

**Cell lines and cell culture.** Human PDAC cell lines PANC-1, BXPC-3, and SW1990 and embryonic kidney cells 293T were purchased from the Cell Bank of Type Culture Collection of the Chinese Academy of Sciences Shanghai Institute of Biochemistry and Cell Biology. Human immortalized pancreatic duct epithelial cell line HPDE6-C7 was purchased from Biotechnology Company. All cell lines were authenticated by the DNA finger printing analysis and tested for free from mycoplasma infection. PANC-1 and 293T cells were maintained in DMEM medium supplemented with 10% FBS while HPDE6-C7, BXPC-3, and SW1990 cells were maintained in RPMI-1640 medium supplemented with 10% FBS. All cell lines were grown without antibiotics in an atmosphere of 5% $CO_2$ and 99% relative humidity at 37 °C.

**Preparation of cigarette smoke condensate.** Cigarette smoke condensate (CSC) was prepared as previously reported[55]. Tobacco smoke was collected from a popular Chinese brand cigarette (12 mg tar per cigarette) by a vacuum machine into a container frozen with liquid nitrogen. CSC was then dissolved in dimethylsulfoxide (DMSO) at a concentration of 235 mg/ml, and aliquots were stored at −80 °C until use. CSC doses used in cell cultivation were 0.1, 1.0, 10, and 100 μg/ml. The maximum dose has been estimated to be relevant to the human exposure situation[56,57].

**MicroRNA array and data analysis.** Total RNA samples for the miRNA microarray analysis were extracted from HPDE6-C7 cells exposed to CSC (100 μg/ml) or equal amount of DMSO as vehicle control for 48 h. The array analysis was performed using the Agilent Human miRNA Microarray (version 21.0). The data and visual heatmap analyses were performed with the MeV 4.4 program (see URLs).

**Quantitative real-time PCR analysis.** Total RNA from the pancreatic tissue specimens and cell lines used in this study was extracted with TRIzol reagent (Invitrogen). Total serum RNA was extracted with the *miR*Vana PARIS kit (Ambion)[58]. Five femtomoles of synthetic cel-miR-54 and cel-miR-39 mixture was added to 100 μl serum and subsequent RNA isolation was performed according to the manufacturer's instructions. First-strand cDNA was synthesized using the RevertAid First Strand cDNA Synthesis Kit (Thermo). Relative RNA levels were determined by qRT-PCR in triplicate on a Roche LightCycler 480 using the SYBR Green method[59]. Beta-actin was employed as an internal control for quantification of the mRNA level of each gene. For miRNA quantification, U6 small nuclear RNA was used as an internal control for tissue and cell lines while cel-miR-54 and cel-miR-39 were used as the control for serum. The relative expression of RNAs was calculated using the comparative Ct method. The primer sequences used for qRT-PCR and other assays in this study are all shown in Supplementary Table 5.

**RNA immunoprecipitation assays.** RNA immunoprecipitation (RIP) assays were performed using the Magna RIP RNA-Binding Protein Immunoprecipitation kit (Millipore). Antibodies from Abcam against Ago2 (ab32381), NKAP (ab121121) or DGCR8 (ab191875) and from Sigma against FLAG (F1804) were used. Total RNA (input control) and isotype control (IgG) for each antibody were assayed simultaneously. The co-precipitated RNAs were detected by qRT-PCR. For determination of the m6A levels in pri-miR-25, total RNA from tissues or cells was extracted and subject to ribosome RNA depletion using the Ribo-Zero Magnetic Kits (Illumina). Ribo-off RNAs were subject to fragmentation using the RNA fragmentation reagents (Ambion). Precipitation was performed using an anti-m6A antibody (Synaptic Systems, 202003) previously bound to magnetic Dynabeads in RIP immunoprecipitation Buffer (Magna RIP Kit, Millipore) and incubated with fragmented RNAs. After treating with proteinase K (10 mg/ml), RNAs was extracted with phenol/chloroform/isoamyl alcohol and subjected to qRT-PCR using the primers for pri-miR-25, which was normalized to input.

**RNA interference.** Small interfering RNA (siRNA) targeting the *METTL3*, *NFIC*, *PHLPP2*, *p70S6K*, or *NKAP* gene and non-targeting siRNA control (Supplementary Table 6) were purchased from GenePharma. Transfections with siRNA (75 nM) were performed with lipofectamine 2000 (Life Technologies).

**Cell lysis and immunoprecipitation**. Cells transfected with DGCR8-6 × His (whole length or truncated constructs) or NKAP-3 × FLAG (whole length or truncated constructs) were lysed with 1 × RIPA buffer supplemented with Protease/Phosphatase Inhibitor Cocktail (Pierce). Lysates were cleared by centrifugation and the supernatants were prepared for immunoblotting or immunoprecipitation with antibodies described below. The supernatants were then treated with RNase A (20 µg/ml) or RNase inhibitor (200 U/ml, New England Biolab) prior to immunoprecipitation. The immunoblot signal was detected using Clarity Western ECL Substrate (Thermo).

**Western blot assays**. Protein extracts from cells or immunoprecipitation samples were prepared using detergent-containing lysis buffer. Total protein (50 µg) was subjected to SDS-PAGE and transferred to PVDF membrane (Millipore). Antibody against METTL3 (ab195352), PHLPP1 (ab71277), PHLPP2 (ab71973), AKT1/2/3 (ab126811), p-AKT1/2/3 (p-S472 + S473 + S474; ab183758), p70S6K (ab32359), p-p70S6K (p-T389; ab126818), NKAP (ab121121), IF4A2 (ab31218), SLU7 (ab151462), PCMD1 (ab121858), PLXA4 (ab127892), CENPJ (ab26052), FLIP1 (ab205925), or DROSHA (ab12286) was from Abcam. Antibody against NFIC (sc-74444), FLAG tag (F1804), 6 × His tag (SAB2702218) or β-ACTIN (66009-1-Ig) was from Santa Cruz Biotechnology, Sigma, or Proteintech, respectively. Membranes were incubated overnight at 4 °C with primary antibody and visualized with a Phototope Horseradish Peroxidase Western Blot Detection kit (Thermo Fisher).

**In vitro proliferation, invasion, and migration assays**. Cells were seeded in 96-well plates (2000 cells per well) and after a certain time of culturation, cell viability was measured using CCK-8 assays (Dojindo). Each experiment with six replicates was repeated three times. Invasion assays were done in a 24-well Millicell chambers in triplicate. The 8-µm pore inserts were coated with 30 µg of Matrigel (BD Biosciences). Cells ($2 \times 10^5$) were added to the coated filters in serum-free medium. We added DMEM medium containing 20% FBS to the lower chambers as a chemoattractant. After 16 h at 37 °C in an incubator at 5% $CO_2$, cells that migrated through the filters were fixed with methanol and stained with crystal violet. Cell numbers in three random fields were counted. The migration assay was conducted in a similar fashion without coating the filters with Matrigel. For testing the effects of CSC exposure on cell phenotypes, cells were incubated with medium supplied with 50 µg/ml of CSC in DMSO or equal amount of DMSO (solvent control) for 1 week before conducting the assays as described above.

**Animal experiments**. Female BALB/c nude mice, aged 4–5 weeks, purchased from the Beijing Vital River Laboratory Animal Technology, were allowed to acclimate to local conditions for 1 week and maintained under a 12-h dark/12-h light cycle with food and water provided ad libitum. Mice (five in each group) were injected subcutaneously with 0.1 ml of cell suspension containing $2 \times 10^6$ cells in the back flank. When a tumor was palpable, it was measured every other day and the volume was calculated according to the formula volume = length × width$^2$ × 0.5. Sample size was not predetermined for these experiments. For metastasis model, 0.1 ml of cell suspension containing $2 \times 10^6$ luciferase labeled cells was injected into tail veins. The metastases were detected using the Living Image® software (Perkin Elmer) 10 min after intraperitoneal injection of 2.0 mg luciferin (Promega) before quantifying fluorescence. All experimenters were blinded to which cells were injected in the mice. All the animal experiments were approved by the Institutional Animal Care and Use Committee of Sun Yat-sen University Cancer Center (reference no. L102022016110Q) and the animals were handled in accordance with institutional guidelines.

**DNA methylation analysis**. The publicly available online tool MethPrimer (see URLs) was used to suggest potential CpG islands in the *METTL3* promoter region with the genomic DNA sequence (1000 nucleotides) before the gene transcription start site. The analysis predicted three potential CpG islands (−50 to −206 bp, −284 to −564 bp, and −680 to −801 bp) within the *METTL3* promoter. Samples for bisulfite-sequencing PCR were conducted according to the protocol. Briefly, genomic DNA from PANC-1 cells treated with CSC (100 µg/ml) or DMSO was extracted using the TIANamp Genomic DNA Kit (DP304, Tiangen); genomic DNA from clinical tissue specimens was extracted using the QIAamp DNA Mini Kit (51304, Qiagene). Purified DNA samples were subsequently subject to bisulfite conversion using the DNA Bisulfite Conversion Kit (DP215, Tiangen). The bisulfite sequencing was commercially accomplished in Sagene Bio (Guangzhou). Methylation-specific PCR was performed using Methylation-Specific PCR kit (EM101, Tiangen) with specific primers (Supplementary Table 6) targeting the CpG island between −284 and −564 bp in the *METTL3* promoter where bisulfite sequencing showed significant difference in the methylation status in cells treated with CSC or DMSO.

**Chromatin immunoprecipitation assays**. Chromatin immunoprecipitation (ChIP) assays were performed using the EZ-Magna ChIP™ A/G Kit (17-10086, Millipore). JASPAR software was used to suggest the potential transcription factors binding the *METTL3* core promoter region as described above. Briefly, cells were cross-linked with 1% formaldehyde, lysed and sonicated on ice to generate DNA fragments with an average length of 200−500 bps. Pre-cleared DNA of each sample was saved as input fraction. Pre-cleared DNA was then used for immunoprecipitation with 5 µg of ChIP-grade antibody specifically against NFIC (sc-74444, Santa

Cruz), DNMT1 (ab13537, Abcam), DNMT3a (ab13888, Abcam), or DNMT3b (Ab13604, Abcam). IgG was included as nonspecific control. DNA was eluted and purified, followed by qRT-PCR using specific primers (Supplementary Table 6).

**In vitro pri-miRNA processing assays**. We transcribed pri-miR-25 in vitro using the T7 based MEGAshortscript kit (Life Technologies). $N^6$-methyl-ATP (m$^6$A) (Biorbit, orb65363) was used instead of ATP in the in vitro transcription reaction to achieve [m$^6$A]pri-miR-25. For pri-miR-25 processing assays, pri-miR-25 or [m$^6$A]pri-miR-25 and pri-miR-1-1 (control) were incubated with whole cellular lysates of 293T cells co-transfected with plasmids carrying DROSHA and DGCR8. Total RNA purified from reaction products was analyzed by northern blotting or qRT-PCR. We also examined the effect of A to T mutation at the m$^6$A site of pri-miR-25 on the processing of pri-miR-25. The mutant pri-miR-25 was in vitro transcribed with the mutant template and in vitro processing was assayed as described above. The primers containing the T7 promoter sequence for in vitro pri-miR-25 transcription are shown in Supplementary Table 6.

**Northern blot assays**. Products from in vitro processing assays were subject to formaldehyde gel electrophoresis and then transferred to a Biodyne Nylon membrane (Pall). The RNA probes labeled with digoxigenin (Supplementary Table 6) were synthesized by Bersinbio. After pre-hybridization for 30 min, the membrane was hybridized for 12 h at 68 °C in buffer containing the denatured probes. After washing, signal on the membrane was detected using an Odyssey infrared scanner (Li-Cor, Lincoln).

**RNA affinity chromatography and mass spectrometry analysis**. RNA pulldown assays were performed as described. Briefly, the biotinylated fragment containing 50-bp pri-miR-25 sequences with or without m$^6$A modified at the splicing site were commercially synthesized and incubated with cellular protein extracts from PANC-1 cells. Streptavidin beads were then added and recovered total proteins were subject to proteomics screening which was accomplished with a mass spectrometry analysis.

**RNA electrophoretic mobility shift assays**. Assays were performed using the LightShift Chemiluminescent RNA EMSA Kit (Life Technologies), with the biotin-labeled RNA oligonucleotides synthesized by Bersinbio. Briefly, 1 µl RNA probes (4 nM final concentration) were incubated in 19 µl binding buffer (10 mM HEPES pH 7.3, 20 mM KCl, 1 mM MgCl$_2$, 1 mM DTT, 5% glycerol, and 40 U/ml RNasin) with different concentrations of recombinant NKAP proteins (as described in the figure legends) at room temperature for 20 min. The RNA-protein mixtures were separated in 8% native polyacrylamide gels (in 0.5 × Tris-borate-EDTA buffer) at 4 °C for 60 min. Complexes were transferred to a nylon membrane, cross-linked to the membrane using the UVP cross-linker (120 mJ/cm$^2$ of 254 nm UV) and detected by chemiluminescence.

**miCLIP-sequencing**. Nuclear RNA from PANC-1 cells was isolated and fragmented. RNA precipitated by ethanol was diluted in 450 µl of RIP buffer and incubated with 15 µg of anti-m$^6$A antibody at 4 °C for 1 h by rotating head over tail. The solution was then transferred into a 12-well cell culture plate and cross-linked twice with 150 mJ/cm$^2$ of UV light (254 nm). After cross-linking, the solution was transferred into Eppendorf tubes and incubated with 100 µl of Protein A/G beads (Millipore) overnight at 4 °C with rotating. Bead-bound antibody-RNA complexes were then recovered on a magnetic stand and washed with high-salt buffer (50 mM Tris, pH 7.4, 1 M NaCl, 1 mM EDTA, 1% NP-40, and 0.1% SDS), immunoprecipitation buffer and polynucleotide kinase (PNK) wash buffer (20 mM Tris, 10 mM MgCl$_2$, and 0.2% Tween 20), respectively. RNA 3′ ends were dephosphorylated on beads with PNK (New England BioLabs) for 30 min in dephosphorylation buffer (70 mM Tris, pH 6.5, 10 mM MgCl$_2$, and 1 mM DTT). After another round of extensive washing (twice with PNK wash buffer, once with immunoprecipitation buffer, once with high-salt buffer, and twice with PNK wash buffer), the 3′ adaptor was ligated with T4 RNA ligase (New England BioLabs) overnight. The antibody-bound RNA was recovered by treatment with proteinase K, acidic phenol/chloroform extraction, and ethanol precipitation. Purified RNA was reverse transcribed with Superscript III reverse transcriptase (Life Technologies). First-strand cDNA was size-selected on an 8% TBE-Urea gel (Life Technologies), and the regions corresponding to 100–180 nucleotides were used for further analysis. After circularization and re-linearization of cDNA, libraries were PCR amplified for 18–21 cycles and sequenced on an Illumina HiSeq X Ten.

**NKAP iCLIP**. In brief, cells overexpressing FLAG-tagged NKAP were washed with ice-cold PBS, cross-linked with 150 mJ/cm$^2$ of 254 nm UV light and harvested on ice. Nuclear extraction was isolated and sonicated, followed by treating with DNase I and low dilution RNase A. Pre-washed Dynabeads protein A/G conjugated with anti-FLAG (F1804, Sigma) antibody were then incubated with the extraction at 4 °C overnight with rotating. RNA was then treated with proteinase K, followed by 3′ linker ligation, acidic phenol/chloroform extraction, and ethanol precipitation. Purified RNA was reverse transcribed with Superscript III reverse transcriptase (Life Technologies) and size-selected on an 8% TBE-Urea gel (Life Technologies).

The regions corresponding to 100–180 nucleotides were used for further analysis. After circularization and re-linearization of cDNA, libraries were PCR amplified for 18–21 cycles and sequenced on an Illumina HiSeq X Ten. For NKAP CLIP qRT-PCR, the input and immunoprecipitated RNA samples were treated with proteinase K, extracted with acidic phenol/chloroform and precipitated with ethanol. cDNA was synthesized with SuperScript III RT and random hexamer primers (Invitrogen) and subject to qRT-PCR using the primers for pri-miR-25.

**Analysis of iCLIP-sequencing data.** Read pre-processing was performed essentially as previously reported[60]. Adaptors and low-quality bases were trimmed by Cutadapt (v1.16), and reads shorter than 20 nucleotides were discarded. Reads were demultiplexed based on their experimental barcode by the pyBarcodeFilter.py script of the pyCRAC tool suite[61]. Sequence-based removal of PCR duplicates was then performed with the pyFastqDuplicateRemover.py script. The reverse reads were reversely complemented and processed in the same way as the forward counterparts. Reads were then mapped to human genome (hg38) with BWA (v0.7.15)[62], with parameter 'bwa aln -n 0.06 -q 20' as recommended by the online CTK Documentation (see URLs). We detected cross-linking-induced mutation sites (CIMS) and cross-linking-induced truncation sites (CITS) in iCLIP data of m6A and NKAP using CLIP Tool Kit (CTK)[63]. To identify the m6A locus, the mode of mutation calling was performed as previously reported[27]. For each mutation position, the coverages of unique tag (k) and mutations (m) were determined by CIMS.pl script of CLIP Tool Kit. First, the known SNPs (dbSNP 147) were removed from all the mutation positions. Sequentially, the C>T mutation positions within m/k ≤ 50% and only mutation positions at the +1 position of adenosines were identified as CIMS-based m6A residues. CITS analysis was performed as recommended by the online CTK Documentation. Truncation sites with a significance value of $p \leq 0.05$ that occurred neighbor adenosines were retained to yield a list of CITS-based m6A residues. For the identification of NKAP-binding sites, the mode of mutation calling was performed as miCLIP with minor modifications. After removal of known SNPs, the mutation positions within FDR ≤ 0.1 were retained and identified as CIMS-based NKAP-binding sites. CITS analysis of NKAP iCLIP was in the same way as described for the miCLIP data. We visualized each iCLIP-seq dataset by counting positional coverage across the genome (bam2wig, see URLs) and viewing in IGV[64].

**Public data processing.** For differential miR-25-3p expression investigation, three miRNA microarray datasets (GSE24279, GSE25820, and GSE41369)[65,66] were downloaded from Gene Expression Omnibus and combined all PAAD and normal pancreatic samples, respectively. Hsa-miR-25-3p expression levels between PDAC ($N = 150$) and normal pancreatic ($N = 39$) samples were compared with unpaired t test. Correlations between the levels of METTL3 and transcript factors were analyzed using data from TCGA and Genotype-Tissues Expression (GTEx) by GEPIA (see URLs). Kaplan–Meier analysis using TCGA data was performed by PROGmiR (see URLs). For HITS-CLIP data processing, we retrieved the published data (GSE61979)[28] for profiling DGCR8-binding sites. We convert genome coordinate of CLIP tags from hg19 to hg38 with CrossMap (v0.2.6)[67]. Piranha (v1.2.1)[68] was applied for peak calling to identify the binding sites of DGCR8.

**Motif enrichment analysis.** Motif enrichment analysis was performed using 21-nucleotide sequences [−10,10] around CIMS or CITS of NKAP iCLIP by AME module of the MEME program[69] with the default parameters.

**Analysis of splicing factor binding sites and splicing sites.** All the splicing binding sites were obtained from the SpliceAid-F database (v1.1 03/2013)[70]. The splicing sites presented were the edge of exon (hg38). We then applied homer (v4.9)[68] to calculate NKAP iCLIP tag density (tag counts per base pair) around the edges of exon.

**In silico analysis of miRNA target genes.** The miRNA target genes were analyzed in silico by using four publicly available algorithms, miRCode, miRDB, miRTar-Base, and TargetScan (see URLs). After overlapping the called targets from each database, common targets were chosen and ranged by confidence scores, which were measured by an algorithm from mirDIP combining the binding energy, evolutionary conservation, ranks, and associated precisions. Higher scores indicate more confidence of the prediction. Target gene predicted with the highest confidence score was chosen for further analysis.

**Construction of vectors.** To construct expression vector for His-tagged DGCR8, synthesized cDNA encoding full-length of DGCR8 or truncated versions of DGCR8 (Obio Technology) were subcloned into pcDNA3.1-6 × His vector. Similarly, to construct expression vector for FLAG-tagged NKAP, synthesized full-length of NKAP cDNA or truncated versions were subcloned into the pcDNA3.1-FLAG vector.

**Plasmids and lentivirus production and transduction.** To construct lentiviral vector expressing human miR-25-3p, full-length pre-miR-25 cDNA was synthesized (Obio Technology) and subcloned into plenty-CMV-IRES-Puromycin

lentiviral expression vector (Obio Technology). To achieve depletion of miR-25-3p in cells, sequence targeting hsa-miR-25-3p (Obio Technology) was subcloned into the same vector. To produce lentivirus containing miR-25-3p or miR-25-3p targeting sequence, 293T cells were co-transfected with the vector described above and the lentiviral vector packaging system (Obio Technology) using lipofectamine 2000. Infectious lentiviruses were collected at 48 and 72 h after transfection and filtered through 0.45-μm PVDF filters. These lentiviruses were, respectively, designated as miR-25-3p overexpression or miR-25-3p knockdown. We used empty plenty-CMV-IRES-Puromycin vector as a negative control. Recombinant lentiviruses were concentrated by centrifugation. The virus-containing pellet was dissolved in DMEM, and aliquots were stored at −80 °C until use. PANC-1, BXPC-3, and SW1990 cells were infected with concentrated virus in the presence of ploybrene (Sigma-Aldrich). The supernatant was replaced with complete culture medium after 24 h, followed by selection with 2 μg/ml puromycin, and the expression of miR-25-3p in infected cells was verified by qRT-PCR.

**Reporter gene assays.** For promoter reporter assays, METTL3 promoter sequence (−284 to −564 bp) was cloned into the pGl3-promoter vector (Promega). For dual-luciferase reporter assays, 500 ng pGl3-METTL3 promoter vector, 100 ng pRL-TK renilla (Promega) luciferase reporter vector, and 50 nM siRNA (i.e., siNC, or siNFIC) were co-transfected into HPDE6-C7 and PDAC cells in 24-well plates. The luciferase activities were assessed 48 h post transfection by Dual-Luciferase Reporter Assay System (Promega) and the relative Fluc/Rluc activity was calculated by normalizing the activity of firefly luciferase to that of renilla luciferase. For microRNA-3'UTR binding assays, a reporter vector in the psiCHECK-2 backbone (Promega) was generated bearing the 3′ untranslated region (UTR) of PHLPP2 (PHLPP2-3'UTR1: 840-bp and PHLPP2-3'UTR: 240-bp). PANC-1 and BXPC-3 cells were seeded in 24-well plates ($1 \times 10^5$ cells/well) and 500 ng of reporter plasmid was co-transfected into cells with 1, 5, or 50 pmol of miR-25-3p mimic (GenePharma) 16 h later using lipofectamine 2000. Cells were collected 24 h after transfection, and luciferase activity was detected using the Dual-Luciferase Reporter Assay System (Promega). Renilla luciferase activity was detected and normalized to firefly luciferase activity.

**Statistical analysis.** For functional analysis, results were presented as mean ± S.D. of three or more experiment repeats. Comparison of mean between two groups was conducted by using Student t-test. Data in abnormal distribution were analyzed by nonparametric test. The $\chi^2$-test was used to examine the relationships between death or miR-25-3p expression and clinicopathological characteristics. Kaplan–Meier methods were used to compare survival by different levels of miR-25-3p expression. Cox proportional hazards models were used to identify independently significant variables. Hazard ratio (HR) and 95% confidence interval (CI) were calculated with age, sex, smoking status, drinking status, and tumor stage as covariates. Spearman's correlations were calculated between miR-25-3p levels and mRNA expression levels of interest genes. Correlations were considered significant and positive when $P < 0.05$ and $r > 0.30$. All statistical analyses were performed using the SPSS software package (version 20.0; IBM SPSS). $P < 0.05$ was considered significant for all statistical analyses.

**URLs.** bam2wig, https://github.com/MikeAxtell/bam2wig; CTK Documentation, https://zhanglab.c2b2.columbia.edu/index.php/CTK_Documentation; JASPAR, http://jaspar.genereg.net/; MethPrimer, http://www.urogene.org/cgi-bin/methprimer/methprimer.cgi; MeV 4.4 program, http://www.tm4.org/mev/; miRCode, http://www.mircode.org/index.php; miRDB, http://mirdb.org/miRDB/; miRTarBase, http://mirtarbase.mbc.nctu.edu.tw/php/search.php; Targetscan, http://www.targetscan.org/vert_71/; GEPIA, http://gepia.cancer-pku.cn/; PROG-miR V2, http://xvm145.jefferson.edu/progmir/index.php; The Cancer Genome Atlas (TCGA), https://cancergenome.nig.gov; The Genotype-Tissue expression project, https://www.gtexportal.org/home/; National Center for Biotechnology Information Gene Expression Omnibus, http://www.ncbi.nlm.nih.gov/geo/.

**Reporting summary.** Further information on experimental design is available in the Nature Research Reporting Summary linked to this article.

## Data availability

The authenticity of this article has been validated by uploading the key raw data onto the Research Data Deposit public platform, with the approval RDD number of RDDB2019000524 (www.researchdata.org.cn). The accession number for the miRNA microarray data is GSE103994 and the miCLIP and NKAP iCLIP data is GSE116425. Unprocessed gel blot of Fig. 2b, e−f, i, 3e, g, 4b, d, e−g, 6d, f−h and 7a, f, Supplementary Figs. 3,a, 7,a and d are available in Supplementary Figs. 8−11. The Source Data underlying Figs. 1b−e, i, 2a−d, g−k, 3b−d, f, h, 4c, h−k, 5c, d, 6a−e, i, j, 7b−e, g−i and Supplementary Figs. 2,a−e, 3,b, 4,a, b, 5b−f, 7b, c are provided as a Source Data file.

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

## Acknowledgements

This study was supported by the Natural Science Foundation of China (81772586 to J. Zheng, 91753142 and 81572793 to D.L., 81602461 to J. Zheng), Program for Guangdong Introducing Innovative and Entrepreneurial Teams (2017ZT07S096), Natural Science Foundation of Guangdong Province (2016A030313283 to J. Zheng), National Young Top-notch Talent Support Program (to J. Zheng), Young Elite Scientists Sponsorship Program by CAST (2017QNRC001 to J. Zheng), Guangdong Province Universities and Colleges Pearl River Scholar Funded Scheme (2017, to J. Zheng), Sun Yat-sen University Intramural Funds (to D.L. and to J. Zheng), and National Postdoctoral Program for Innovative Talent (BX20180392 to J. Zhang).

## Author contributions

D.L., J. Zheng, R.X., and R.C. conceptualized and supervised the research. J. Zhang, R.B., and M.L. contributed to the study design and performed most of assays. D.M., L.T., L.P., H.Y., and G.L. performed animal experiments. Y.Z., J.S., Y.Y., Z.Z., Z.L., and Q.Z. were engaged in biostatistics and bioinformatics analyses. S.L., C. Wu, C. Wang, W.T., X.C., Z.F., S.Z., D.X., W.J., and M.Z. contributed to collection of clinic samples and analysis of clinical data. D.L., J. Zheng, and J. Zhang prepared manuscript.

## Additional information

**Competing interests:** The authors declare no competing interests.

