## [Peer Review File · Nature Communications]

Reviewers' comments:

Reviewer #1 (Remarks to the Author):

It has been well established that cigarette smoking is a risk factor for PDAC, but the underlying mechanism remains largely unknown. In this manuscript, Zhang et al propose a mechanistic link between CSC and PDAC by showing that CSC may cause the dysregulation of miR-25-3p via aberrant adenosine methylation (m6A) of its primary microRNA. They first conducted microarray analysis of the differential expression of microRNAs in an immortalized human pancreatic duct epithelial cell treated with CSC and found that miR-25-3p was the most dramatically upregulated miRNA upon exposure to CSC. They then found that the miR-25-3p levels were significantly higher in PDAC than in normal pancreatic tissues, which is significantly correlated with poor survival in PDAC patients. Furthermore, they detected higher serum miR-25-3p in smokers compared with nonsmokers. They found that CSC may cause demethylation in the METTL3 promoter, facilitating the binding of transcription factor NFIC to its regulatory region and consequently enhanced expression of METTL3 in CSC-exposed PDAC cells. They also investigated the mechanism by which m6A methylation promotes miR-25-3p maturation and found that NKAP may mediate DGCR8 recognition of pri-miR-25. They also identified PHLPP2 as the downstream target of miR-25-3p. High level of miR-25-3p suppresses PHLPP2, then activates the downstream ATK-p70S6K signaling pathway. Overall, this is a well-designed comprehensive study illustrating several novel findings, whereas the following issues need to be addressed by the authors.

1. Figure 2c, 3d and 5d can be more convincing with appropriate stats.
2. The author should address the regulation of METTL3 promoter by NFIC via in vitro assay, such as reporter assay, to further support the result that binding of NFIC to METTL3 promoter is truly regulated by methylation status of its regulatory region.
3. In Supplementary Figure 4b, sample count should be addressed in the figure legend.
4. The current version of manuscript contains a wealth of data that somehow makes the manuscript less focusing. As such, the data section needs to be streamlined and some of the less important supplementary information (e.g. supplementary data 4) are recommended to be omitted, which would certainly render the paper clearer and more digestible for the readers.
5. The current version of manuscript is not well-written, with some grammatical errors, and inappropriate descriptions (e.g. "remarkably" & "dramatic" etc). More objective descriptions where appropriate should be used when presenting the results.
6. Updated references about the important roles of m6A in cancer should be included.
7. Many chemicals exist in CSC, such as pro-carcinogens, neurotransmitters, etc. What product(s) in tobacco smoke causes the phenotypes described here?

Reviewer #2 (Remarks to the Author):

This manuscript has adequately addressed my concerns from a prior review. I find the manuscript suitable for publication in Nature Communications.

Reviewer #3 (Remarks to the Author):

Zhang et al reported here that cigarette smoke increases m6A modification within pri-miR-25 by

METTL3 which, in turn, promotes its maturation by DROSHA, in pancreatic cancer. They go on showing that Increase in DROSHA processing augments miR-25 level, which is an oncogenic miRNA in pancreatic cancer because directly represses PHLPP2. Interestingly they further found that a novel m6A reader, NKAP, interacts with m6A modification presents on pri-miR-25 which induces DGCR8 and DROSHA recruitment on this primary miRNA.

This study reports several novels, interesting findings, therefore I believe that, after some extra experimental work, it deserves to be published in Nature Communications.

Specifically:

1) In figure 1e the author reported that miR-25-3p is up-regulated in PDAC compared to normal samples in two independent cohorts. These results are not totally convincing because of the evident high variability in miR-25-3p expression across the samples. However, several miRNA expression profiling studies of miRNA levels in PDAC comparing to normal specimens have been previously reported.

The authors should perform a metanalysis of these data to verify whether miR-25-3p is recurrently over-expressed in PDAC by exploiting these multiple datasets.

Also, what is the relationship between miR-25-3p expression and PDAC grade?

2) In order to improve survival data (fig 1f-h), it would be interesting to also include analysis of publicly available datasets of miRNA expression in PDAC, such as the ones coming from TCGA data.

3) The authors indicate that cigarette smoke concentrate (CSC) increases miR-25 maturation through NFIC, METTL3 and NKAP, but they do not properly demonstrate this model. To do so, the authors should test whether CSC-dependent increase of miR-25-3p maturation can be rescued by siRNA treatment of NFIC, METTL3 and/or NKAP.

4) Methods lack information regarding correlation analyses of gene pair obtained by using TCGA or GTEx datasets and should be included in the revised version.

RESPONSES TO THE REVIEWERS' COMMENTS

Reviewer #1 (Remarks to the Author):

It has been well established that cigarette smoking is a risk factor for PDAC, but the underlying mechanism remains largely unknown. In this manuscript, Zhang et al propose a mechanistic link between CSC and PDAC by showing that CSC may cause the dysregulation of miR-25-3p via aberrant adenosine methylation (m⁶A) of its primary microRNA. They first conducted microarray analysis of the differential expression of microRNAs in an immortalized human pancreatic duct epithelial cell treated with CSC and found that miR-25-3p was the most dramatically upregulated miRNA upon exposure to CSC. They then found that the miR-25-3p levels were significantly higher in PDAC than in normal pancreatic tissues, which is significantly correlated with poor survival in PDAC patients. Furthermore, they detected higher serum miR-25-3p in smokers compared with nonsmokers. They found that CSC may cause demethylation in the METTL3 promoter, facilitating the binding of transcription factor NFIC to its regulatory region and consequently enhanced expression of METTL3 in CSC-exposed PDAC cells. They also investigated the mechanism by which m⁶A methylation promotes miR-25-3p maturation and found that NKAP may mediate DGCR8 recognition of pri-miR-25. They also identified PHLPP2 as the downstream target of miR-25-3p. High level of miR-25-3p suppresses PHLPP2, then activates the downstream ATK-p70S6K signaling pathway. Overall, this is a well-designed comprehensive study illustrating several novel findings, whereas the following issues need to be addressed by the authors.

Response: We thank the Reviewer for this very positive overview.

Comment 1: Figure 2c, 3d and 5d can be more convincing with appropriate stats.

Response 1: Per the suggestion, we have added appropriate stats to Figs 2c, 3d and 5d in the revision.

Comment 2: The author should address the regulation of METTL3 promoter by NFIC via in vitro assay, such as reporter assay, to further support the result that binding of NFIC to METTL3 promoter is truly regulated by methylation status of its regulatory region.

Response 2: We have performed luciferase reporter assays addressing the directly regulation of *METTL3* promoter by NFIC and the results support the role of NFIC on regulating *METTL3* expression on transcriptional level. We have added these results in the revision (page 7, lines 167–169; page 27–28, lines 654–660; Supplementary Fig. 5c and its legends).

Comment 3: In Supplementary Figure 4b, sample count should be addressed in the figure legend.

Response 3: Sample count in Supplementary Fig. 4b has been added to the figure legend. This figure is designated as Supplementary Fig. 5b in the revision (Supplementary Fig. 5b legend).

Comment 4: The current version of manuscript contains a wealth of data that somehow makes the manuscript less focusing. As such, the data section needs to be streamlined and some of the less important supplementary information (e.g. supplementary data 4) are recommended to be omitted, which would certainly render the paper clearer and more digestible for the readers.

Response 4: Thanks for the comment. We have removed Supplementary data 4 in the revision.

Comment 5: The current version of manuscript is not well-written, with some grammatical errors, and inappropriate descriptions (e.g. “remarkably” & “dramatic” etc). More objective descriptions where appropriate should be used when presenting the results.

Response 5: We have carefully edited out grammatical errors and replaced inappropriate descriptions such as remarkably and dramatic with appropriate description.

Comment 6: Updated references about the important roles of m⁶A in cancer should be included.

Response 6: Per the comment, we have added references 41 and 45–49, which have been published recently demonstrating the important roles of m⁶A in cancer.

Comment 7: Many chemicals exist in CSC, such as pro-carcinogens, neurotransmitters, etc. What product(s) in tobacco smoke causes the phenotypes described here?

Response 7: Thanks for bringing up this comment. Indeed, CSC contains many kinds of chemicals including pro-carcinogens, neurotransmitters, etc. In the present study, we showed CSC induces miR-25-3p excessive maturation via m⁶A modification and it might play a role in the development and progression of pancreatic cancer. However, at this stage, we are not clear what component(s) in CSC cause the phenotypes. It has been shown that even DNA-damaging pro-carcinogens, such as PAHs, can cause epigenetic alternations in human (Li et al, *Environ Health Perspect* 2018, doi/10.1289/EHP2773). It would be interesting to find out what product(s) in tobacco smoke causes the phenotypes in the future despite the work is of great challenge because of too many chemicals.

Reviewer #2 (Remarks to the Author):

This manuscript has adequately addressed my concerns from a prior review. I find the manuscript suitable for publication in Nature Communications.

Response: The comment of this Reviewer is every positive and no response is needed.

Reviewer #3 (Remarks to the Author):

Zhang et al reported here that cigarette smoke increases m⁶A modification within pri-miR-25 by METTL3 which, in turn, promotes its maturation by DROSHA, in pancreatic cancer. They go on showing that Increase in DROSHA processing augments miR-25 level, which is an oncogenic miRNA in pancreatic cancer because directly represses PHLPP2. Interestingly they further found that a novel m⁶A reader, NKAP, interacts with m⁶A modification presents on pri-miR-25 which induces DGCR8 and DROSHA recruitment on this primary miRNA.

This study reports several novels, interesting findings, therefore I believe that, after some extra experimental work, it deserves to be published in Nature Communications.

Response: Many thanks for the positive comments and encouragement.

Comment 1: In figure 1e the author reported that miR-25-3p is up-regulated in PDAC compared to normal samples in two independent cohorts. These results are not totally convincing because of the evident high variability in miR-25-3p expression across the samples. However, several miRNA

expression profiling studies of miRNA levels in PDAC comparing to normal specimens have been previously reported. The authors should perform a meta-analysis of these data to verify whether miR-25-3p is recurrently over-expressed in PDAC by exploiting these multiple datasets. Also, what is the relationship between miR-25-3p expression and PDAC grade?

Response 1: As per the suggestion, we have analyzed the miRNA expression profiling in PDAC in Gene Expression Omnibus database (GSE24279, GSE25820 and GSE41369) and the results show that PDACs expressed higher miR-25-3p level than normal pancreatic tissues. We have added this result in revised manuscript (Supplementary Fig. 1, page 5, lines 118–120). Besides, previous study reported by other group also showed that miR-25-3p was significantly up-regulated in pancreatic cancer compared with normal pancreatic tissue (Volinia et al, Proc. Natl. Acad. Sci. USA 23, 2152–2165, 2006, doi/10.1073/ pnas.0510565103). We have included this information in revised manuscript (Reference 36). These previous findings are consistent with our result shown in Fig. 1e.

We have analyzed the relationship between miR-25-3p expression levels and PDAC stage in our patients' cohorts but the results are negative. Two reasons might explain the negative result: first, although we have two independent patient groups in this study, the sample sizes ($N=102$ and $N=73$, respectively) are relatively small for further stratification analysis of grade or stage; second, to obtain tissue specimens for molecular analysis, 83.3% of patients in the present study were in early stage (I/II) and underwent surgical resection, which may have selection bias for the relationship analysis. Further studies with randomly selected larger sample size are needed to address this issue.

We also analyzed the TCGA data for differential cancer-normal expression of miR-25-3p and relationship between miR-25-3p expression levels and PDAC stage. Unfortunately, there are only 4 paired PDAC and normal tissue samples in TCGA database (Fig. 1a for Reviewer), which cannot provide conclusive result. However, we found an increasing trend for miR-25-3p levels in terms of increasing PDAC stages (Fig. 1b for Reviewer) although the differences were not statistically significant. Because this is primarily negative results, we did not include it in the manuscript. One previous study has reported that level of miR-25 gradually increased in multistep tumorigenesis of pancreatic cancer, ranging from normal, hyperplastic, angiogenic islets, to tumors, indicating a high correlation between miR-25 expression and PDAC stages (Olson et al. *Genes Dev.* 23, 2152–2165, 2009, doi/10.1101/gad.1820109). We have added this reference in the revision (Reference 35).

Fig. 1 for Reviewer The expression levels of miR-25-3p in PDAC from TCGA database. **a** Expression level of miR-25-3p was detected in 4 paired PDAC and normal tissue samples in TCGA database. **b** Shown were increasing trend for miR-25-3p levels in terms of increasing PDAC T and N stages.

Comment 2: In order to improve survival data (fig 1f-h), it would be interesting to also include analysis of publicly available datasets of miRNA expression in PDAC, such as the ones coming from TCGA data.

Response 2: As per the suggestion, we have analyzed the TCGA database for the survival issue. No positive result is seen for overall survival of PDAC patients (Fig. 2 for Reviewer, *left*). We found that patients with high level of miR-25-3p had worse relapse-free survival compared with those with low level of miR-25-3p (Fig. 2 for Reviewer, *right*), but this is not statistically significant. The reason for the inconsistency between our results and the TCGA data is not evident, probably due to differences in other molecular or genetic alterations between different races of patients in study. Because the result coming from TCGA data is basically negative, we did not include this information in the manuscript.

Fig. 2 for Reviewer The association between miR-25-3p level and survival of PDAC patients from TCGA database.

Comment 3: The authors indicate that cigarette smoke concentrate (CSC) increases miR-25 maturation through NFIC, METTL3 and NKAP, but they do not properly demonstrate this model. To do so, the authors should test whether CSC-dependent increase of miR-25-3p maturation can be rescued by siRNA treatment of NFIC, METTL3 and/or NKAP.

Response 3: In the initial manuscript, we did test the effects of siRNA treatment of METTL3 or NKAP on CSC-induced miR-25-3p maturation and the results are positive (Fig. 2d for METTL3 and Fig. 4k for NKAP). Per the suggestion, we have performed additional experiments to test the effects of siRNA treatment of NFIC on CSC-induced miR-25-3p maturation and the result is also positive and is in line with our conclusion. We have included this new result in the revision (Page 7, lines 171–172; Supplementary Fig. 5f). We hope that the addition of this result has satisfied the comment.

Comment 4: Methods lack information regarding correlation analyses of gene pair obtained by using TCGA or GTEx datasets and should be included in the revised version.

Response 4: Thanks. Per the comment, we have added the information on the analysis methods in the public data processing section under Methods (Pages 25–26, lines 607–612) and the websites of corresponding database or platform are also included in the “URLs” part (Page 29).

REVIEWERS' COMMENTS:

Reviewer #1 (Remarks to the Author):

The authors have now addressed my comments. I am now pleased to recommend it for publication.

Reviewer #3 (Remarks to the Author):

The authors answered the most of my requests sufficiently. However, I do not agree with the authors that TCGA survival data for miR-25 should not be included in the manuscript because negative.

I suggest the authors to insert this results in the supplementary figure section and include convincing speculations on the reason why TCGA results are negative within the discussion.

RESPONSES TO THE REVIEWERS' COMMENTS

Reviewer #1 (Remarks to the Author):

The authors have now addressed my comments. I am now pleased to recommend it for publication.

Response: Thanks. No response to Reviewer #1 is needed.

Reviewer #3 (Remarks to the Author):

The authors answered the most of my requests sufficiently.

However, I do not agree with the authors that TCGA survival data for miR-25 should not be included in the manuscript because negative.

I suggest the authors to insert this result in the supplementary figure section and include convincing speculations on the reason why TCGA results are negative within the discussion.

Response: Per the suggestion, we have added the TCGA survival data for miR-25 in revised manuscript (lines 126 and 127; Supplementary Fig. 1b). The reason why the TCGA survival result is not consistent with ours in the present study is not clear. However, we speculate that this might be due to the differences in other genetic or epigenetic alterations between different races of patients in study. We have included this speculation directly after the statement of result (lines 127 and 128). We hope this revision satisfies the Reviewer.